# Improved Bounds for Private and Robust Alignment

Wenqian Weng [* 1]   Yi He [* 1]   Xingyu Zhou [1]

## Abstract

In this paper, we study the private and robust alignment of language models from a theoretical perspective by establishing upper bounds on the suboptimality gap in both offline and online settings. We consider preference labels subject to privacy constraints and/or adversarial corruption, and analyze two distinct interplays between them: *privacy-first* and *corruption-first*. For the privacy-only setting, we show that log loss with an MLE-style algorithm achieves near-optimal rates, in contrast to conventional wisdom. For the joint privacy-and-corruption setting, we first demonstrate that existing offline algorithms in fact provide stronger guarantees—simultaneously in terms of corruption level and privacy parameters—than previously known, which further yields improved bounds in the corruption-only regime. In addition, we also present the first set of results for private and robust *online* alignment. Our results are enabled by new uniform convergence guarantees for log loss and square loss under privacy and corruption, which we believe have broad applicability across learning theory and statistics.

## 1. Introduction

The alignment post-training in large language models (LLMs) relies on human preference data to ensure that model outputs are both helpful and harmless (Bai et al., 2022; Ouyang et al., 2022). While widely adopted in practice, these preference labels are often noisy (Lambert et al., 2023). Noise can arise from corruption or misspecification during label generation—such as data poisoning attacks (Casper et al., 2023)—or from privacy concerns that lead individuals to provide noisy and privatized preferences rather than their true rankings (Feng et al., 2024).

Motivated by these challenges, recent works have begun to study the theoretical impact of noisy preference labels in alignment, e.g., (Zhou et al., 2025b;a; Chowdhury et al., 2024b;a). However, several aspects of the existing results remain unsatisfactory.

First, on the privacy side, current approaches (Zhou et al., 2025b;a; Chowdhury et al., 2024b;a) typically require designing new losses that deviate from the natural maximum-likelihood-estimation (MLE) loss. Some even suggest that the private MLE log loss fails to achieve a near-optimal rate (Chowdhury et al., 2024b).

> **Q1.** Can we achieve a near-optimal rate for private alignment using just MLE-type loss?

**Contribution 1.** We give an affirmative answer: the standard private MLE-type log loss can, in fact, achieve a near-optimal rate. This sharpens our understanding of private alignment. Our result is enabled by a new uniform convergence result for log loss under label privacy, an ingredient that may be useful more broadly in other learning scenarios.

Turning to adversarial Huber corruption and its interplay with privacy, existing state-of-the-art guarantees are strictly suboptimal (Zhou et al., 2025b).

> **Q2.** Can we achieve improved rates for private and robust alignment?

**Contribution 2.** We give an affirmative answer to **Q2**: through a new analysis, we show that existing algorithms already admit stronger guarantees in both the corruption-only and the joint corruption–privacy settings. This improvement is enabled by a new uniform convergence result for the square loss under privacy and corruption, which is widely useful given the square loss's central role in reinforcement learning broadly (Agarwal et al., 2019).

Finally, most prior results on private and robust alignment focus on the offline setting, where guarantees inevitably depend on how well the dataset covers the optimal policy (Huang et al., 2024). By contrast, recent advances show that online alignment with active exploration can achieve rates determined only by the intrinsic complexity of the policy class (Xie et al., 2024).

---

[*]Equal contribution [1]Department of Electrical and Computer Engineering, Wayne State University, Detroit, USA. Correspondence to: Xingyu Zhou <xingyu.zhou@wayne.edu>.

*Proceedings of the 43rd International Conference on Machine Learning*, Seoul, South Korea. PMLR 306, 2026. Copyright 2026 by the author(s).

*Table 1.* Summary of theoretical guarantees in the *offline* alignment setting. Here $\kappa(\pi^\star)$ denotes the single-policy concentrability coefficient.

| Method | Loss | Scenario | Theoretical Bound |
|---|---|---|---|
| Square$\chi$PO (Zhou et al., 2025b) | Square | Private + Corruption | CTL: $O\left(\kappa(\pi^\star)\left(c(\varepsilon)\sqrt{\frac{\log(\|\Pi\|/\delta)}{n}} + \sqrt{\alpha}\right)\right)$ 
 LTC: $O\left(\kappa(\pi^\star)\left(c(\varepsilon)\sqrt{\frac{\log(\|\Pi\|/\delta)}{n}} + \sqrt{c(\varepsilon)\alpha}\right)\right)$ |
| Square$\chi$PO (Ours) | Square | Private + Corruption | CTL: $O\left(\kappa(\pi^\star)\left(c(\varepsilon)\sqrt{\frac{\log(\|\Pi\|/\delta)}{n}} + \alpha\right)\right)$ 
 LTC: $O\left(\kappa(\pi^\star)\left(c(\varepsilon)\sqrt{\frac{\log(\|\Pi\|/\delta)}{n}} + c(\varepsilon)\alpha\right)\right)$ |
| Priv$\chi$PO (Ours) | Log (MLE) | Private only | $O\left(\kappa(\pi^\star)\,c(\varepsilon)\sqrt{\frac{\log(\|\Pi\|/\delta)}{n}}\right)$ |

*Table 2.* Summary of theoretical guarantees in the *online* alignment setting. Here $\kappa_{\mathrm{cov}}(\Pi)$ denotes the coverability coefficient.

| Method | Loss | Scenario | Theoretical Bound |
|---|---|---|---|
| XPO (Xie et al., 2024) | Log (MLE) | Non-private | $O\left(\kappa_{\mathrm{cov}}(\Pi)\sqrt{\frac{\log(\|\Pi\|T)}{T}}\right)$ |
| SquareXPO (Ours) | Square | Private + Corruption | CTL: $O\left(\kappa_{\mathrm{cov}}(\Pi)\left(c(\varepsilon)\sqrt{\frac{\log(\|\Pi\|T/\delta)}{T}} + \alpha\right)\right)$ 
 LTC: $O\left(\kappa_{\mathrm{cov}}(\Pi)\left(c(\varepsilon)\sqrt{\frac{\log(\|\Pi\|T/\delta)}{T}} + c(\varepsilon)\alpha\right)\right)$ |
| PrivXPO (Ours) | Log (MLE) | Private only | $O\left(\kappa_{\mathrm{cov}}(\Pi)\,c(\varepsilon)\sqrt{\frac{\log(\|\Pi\|T/\delta)}{T}}\right)$ |

**Q3.** Can we obtain theoretical guarantees for online private and/or robust alignment?

**Contribution 3.** The answer is yes. Building on our new uniform convergence results for log and square losses under privacy and corruption, we show that a simple one-line modification of existing online alignment algorithms suffices to achieve near-optimal rates in the presence of noisy preference labels.

Taken together, our results advance the theoretical foundations of private and robust alignment post-training and open the door to broader applications of uniform convergence in other learning scenarios.

Tables 1 and 2 summarize our theoretical guarantees relative to prior work in the offline and online settings, respectively. We further provide empirical evaluations in Appendix B, including experiment on the real-world PKU-SafeRLHF preference dataset.

## 2. Preliminaries

In this section, we present some background on the alignment in language models as well as its privacy and robustness issues.

### 2.1. Offline and Online Alignment

In the offline setting (Ouyang et al., 2022), the learner has access to an *offline preference* dataset $\mathcal{D}_{\mathsf{pref}} = \{(\tau_{-1}^{(i)}, \tau_1^{(i)}, y^{(i)})\}_{i=1}^n$, which consists of two responses and one preference label. The two responses (full trajectories) are i.i.d. samples from a reference policy $\pi_{\mathsf{ref}}$ starting from an initial prompt (state) $s^{(i)} \sim \rho$, i.e., $\tau_{-1}^{(i)} \sim \pi_{\mathsf{ref}} \mid s^{(i)}$ and $\tau_1^{(i)} \sim \pi_{\mathsf{ref}} \mid s^{(i)}$. We assume that each response includes the initial state. The preference label $y^{(i)} \in \{-1, 1\}$ is generated according to $\{-1, 1\}$-valued Bernoulli distribution, i.e., $y^{(i)} \sim \mathsf{Ber}(\mathcal{P}^\star(\tau_1^{(i)} \succ \tau_{-1}^{(i)} \mid s^{(i)}))$, where $\mathcal{P}^\star(\tau_1^{(i)} \succ \tau_{-1}^{(i)} \mid s^{(i)}) \in [0, 1]$ is the probability that given $s^{(i)}$, $\tau_1^{(i)}$ is preferred over $\tau_{-1}^{(i)}$. Following the literature, we assume that the preference follows the *Bradley-Terry* model (Bradley & Terry, 1952): for two responses $\tau$ and $\tau'$ from $s$

$$\mathcal{P}^\star(\tau \succ \tau' \mid s) = \frac{\exp(r(\tau))}{\exp(r(\tau)) + \exp(r(\tau'))}, \quad (1)$$

where the unknown function $r \in [0, R_{\mathsf{max}}]$ specifies the reward for each response. The goal in offline alignment is often to find a policy that maximizes the (unregularized) reward

$$J(\pi) := \mathbb{E}_\pi[r(\tau)],$$

where $\mathbb{E}_\pi[\cdot] := \mathbb{E}_{s\sim\rho,\tau\sim\pi(\cdot|s)}[\cdot]$.

In the online setting, the learner has access to *online samples* using the current model during training (Guo et al., 2024). In particular, the protocol proceeds in $T$ rounds. At each round $t$, the learner generates two responses $\tau^{(t)}, \widetilde{\tau}^{(t)}$ using the current policy $\pi^{(t)}$ and some other sampling policy, respectively, starting from a sampled initial state $s^{(t)}$ from a distribution $\rho$.

The two responses are then labeled as $y^{(t)} \in \{-1, 1\}$ via human feedback, such that if $y^{(t)} = 1$, $\tau^{(t)}$ is preferred over $\widetilde{\tau}^{(t)}$ and if $y^{(t)} = -1$, $\widetilde{\tau}^{(t)}$ is preferred over $\tau^{(t)}$. As before, we assume the preference follows from the Bradley-Terry model as in (1). Then, the online preference dataset is updated via $\mathcal{D}_{\mathsf{pref}}^{(t)} = \mathcal{D}_{\mathsf{pref}}^{(t-1)} \cup \{(\tau^{(t)}, \widetilde{\tau}^{(t)}, y^{(t)})\}$. The goal in online alignment is to leverage the online interaction to find a policy that maximizes the (regularized) reward

$$J_\beta(\pi) := \mathbb{E}_\pi\left[r(\tau) - \beta \log \frac{\pi(\tau)}{\pi_{\mathsf{ref}}(\tau)}\right], \qquad (2)$$

where $\beta > 0$ is a parameter for the KL regularization.

*Remark* 2.1 (Unregularized vs. regularized objective). For a fair comparison, our learning objectives for offline and online alignment follow from the existing state-of-the-art results (Huang et al., 2024; Xie et al., 2024). We also note that one can often convert one guarantee in the regularized objective to the unregularized one, under certain conditions, and vice versa.

## 2.2. Privacy and Robustness in Alignment

A natural privacy concern in the alignment process is that the preference label could reveal humans' private and sensitive information. Thus, for privacy protection, the learner may only have access to privatized labels, which is achieved by a local randomizer that randomly flips the label. This can ensure the so-called *local differential privacy* (LDP) (Warner, 1965; Kasiviswanathan et al., 2011; Duchi et al., 2013).

**Definition 2.2** (Randomized response and $\varepsilon$-LDP (Warner, 1965)). Let $\varepsilon > 0$ be the privacy parameter and $y \in \{-1, 1\}$ be the true label. The randomized response (RR) mechanism $\mathcal{R}$ flips $y$ and outputs private $\widetilde{y} \in \{-1, 1\}$ based on the following distribution

$$\mathbb{P}\left[\widetilde{y} = y\right] = \frac{e^\varepsilon}{1+e^\varepsilon} \text{ and } \mathbb{P}\left[\widetilde{y} \neq y\right] = \frac{1}{1+e^\varepsilon}. \qquad (3)$$

This can be easily shown to satisfy $\varepsilon$-LDP, i.e., for any $y, y'$ and any subset $S$ in the range of $\mathcal{R}$ such that

$$\mathbb{P}[\mathcal{R}(y) \in S] \leq e^\varepsilon \cdot \mathbb{P}\left[\mathcal{R}\left(y'\right) \in S\right].$$

*Remark* 2.3. In this paper, we mainly focused on the randomized response mechanism specifically, rather than a general

setup where LDP is satisfied. A short reason is that randomized response has already allowed us to achieve the minimax lower bound, since all the lower bounds are general lower bounds for any LDP mechanisms rather than only for randomized response. Another reason here is that any $\varepsilon$-LDP mechanism over $\{-1, 1\}$ can be reduced to a randomized response with two properly chosen distributions (Cheu et al., 2019).

Moreover, with respect to the preference labels, once the RR mechanism has produced the privatized labels, all subsequent training and policy-selection procedures are post-processing of the RR outputs and therefore preserve the same $\varepsilon$-LDP guarantee, provided that they do not access the raw labels.

In practice, the preference labels may be corrupted due to adversary manipulation or simple recording errors. To capture this, we borrow the classic *Huber corruption* model from robust statistics.

**Definition 2.4** ($\alpha$-Huber corruption (Huber, 1964)). We consider the following $\alpha$-Huber corruption: let $\alpha \in [0, 1/2)$, the corrupted label is independently sampled from the mixture $(1-\alpha)G + \alpha B$, where $G$ is the clean/true Bernoulli distribution of the label, and $B$ is some arbitrary unknown $\{-1, 1\}$-valued Bernoulli distribution. That is, with probability $\alpha$, the observed label is sampled from some bad distribution.

As in previous work (Zhou et al., 2025b), we mainly focus on the interplay between privacy and corruption by considering the co-existence of them with different orders, which gives results for the corruption-only case.

**Definition 2.5** (CTL and LTC). Let $\alpha \in [0, 1/2), \varepsilon > 0$. We consider the following two different interplays:

*Corruption-then-LDP* (CTL). The raw (offline or online) label is first corrupted by the $\alpha$-Huber model, which is then further privatized by $\varepsilon$-LDP RR mechanism. This gives the final offline preference dataset $\bar{\mathcal{D}}_{\mathsf{pref}} = \{\tau_{-1}^{(i)}, \tau_1^{(i)}, z^{(i)}\}_{i=1}^n$ or online preference dataset $\bar{\mathcal{D}}_{\mathsf{pref}}^{(t)} = \{(\tau^{(i)}, \widetilde{\tau}^{(i)}, z^{(i)})\}_{i=1}^t$ for each $t \in [T]$.

*LDP-then-Corruption* (LTC). The raw (offline or online) label is first privatized by $\varepsilon$-LDP RR mechanism, which is then further corrupted by the $\alpha$-Huber model. This gives the final offline preference dataset $\bar{\mathcal{D}}_{\mathsf{pref}} = \{\tau_{-1}^{(i)}, \tau_1^{(i)}, z^{(i)}\}_{i=1}^n$ or online preference dataset $\bar{\mathcal{D}}_{\mathsf{pref}}^{(t)} = \{(\tau^{(i)}, \widetilde{\tau}^{(i)}, z^{(i)})\}_{i=1}^t$ for each $t \in [T]$.

*Remark* 2.6. Given the above definition, several comments are in order. First, we do not distinguish between the final observed datasets under CTL and LTC, since our later proposed algorithm does not require knowing the specific setting in advance, since it is agnostic to the corruption order. Second, in the online setting, we consider an oblivious

adversary, where the choice of the bad distribution at round $t$ is independent of other samples. Third, since the label LDP is essentially a random flipping noise/corruption, we can also view CTL and LTC as the interplay between random flipping corruption and adversary corruption. In the same sense, our results for the private case naturally hold for the scenario with random flipping corruption as in Chowdhury et al. (2024a).

## 3. Uniform Convergence: Privacy and Corruption

In this section, we will present our main technical contributions: uniform convergence results for both log loss and square loss under privacy and/or corruption.

### 3.1. Log Loss under LDP

In this subsection, our main result is the following lemma, which gives the uniform convergence with log loss under local privacy. This can be viewed as a variant of the standard MLE-type generalization guarantee under log loss (Van de Geer, 2000).

**Lemma 3.1.** *Suppose that* $\{P_\theta(y|x)\}_{\theta \in \Theta} \subseteq (\mathcal{X} \to \Delta(\{-1, 1\}))$ *is a class of conditional densities parameterized by a finite class* $\Theta$. *Consider the clean data* $(x^{(1)}, y^{(1)}), \ldots, (x^{(T)}, y^{(T)})$ *be a sequence of random variables adapted to a filtration* $(\mathcal{F}^{(t)})_{t=1}^T$ *such that realizability holds, i.e., there exists* $\theta^\star \in \Theta$ *so that* $\mathbb{P}(y^t = \cdot | x^{(t)}, \mathcal{F}^{(t-1)}) = P_{\theta^\star}(y^{(t)} = \cdot | x^{(t)})$ *almost surely for* $t \in [T]$, *and the privatized data be* $(x^{(1)}, \widetilde{y}^{(1)}), \ldots, (x^{(T)}, \widetilde{y}^{(T)})$ *obtained by randomized response mechanism on the labels with parameter* $\varepsilon > 0$ *(cf. (3)). Then, with probability at least* $1 - \delta$, *it holds that for* $n \in [T]$, *for all* $\theta \in \Theta$,

$$\sum_{t=1}^n \mathbb{E}\left[ \left\| P_\theta(\cdot|x^{(t)}) - P_{\theta^\star}(\cdot|x^{(t)}) \right\|_{\mathsf{TV}}^2 \mid \mathcal{F}^{(t-1)} \right]$$
$$\lesssim c(\varepsilon)^2 \cdot \left( \widehat{L}^{(n)}(\theta) - \widehat{L}^{(n)}(\theta^\star) + \log(|\Theta|\delta^{-1}) \right),$$

*where* $a \lesssim b$ *is a shorthand for* $a = \mathcal{O}(b)$, $c(\varepsilon) := \frac{e^\varepsilon + 1}{e^\varepsilon - 1} = \frac{1}{2\sigma(\varepsilon) - 1}$ *and for any given* $\theta' \in \Theta$, $\widehat{L}^{(n)}(\theta') := \sum_{t=1}^n -\log \widetilde{P}_{\theta'}(\widetilde{y}^{(t)}|x^{(t)})$ *with* $\widetilde{P}_{\theta'}(\widetilde{y}^{(t)}|x^{(t)}) := \sigma(\varepsilon) \cdot P_{\theta'}(\widetilde{y}^{(t)}|x^{(t)}) + (1 - \sigma(\varepsilon)) \cdot P_{\theta'}(-\widetilde{y}^{(t)}|x^{(t)}) = (2\sigma(\varepsilon) - 1)P_{\theta'}(\widetilde{y}^{(t)}|x^{(t)}) + (1 - \sigma(\varepsilon))$ *being the private probability.*

*Remark* 3.2. For the ease of presentation, we focus on the finite class case. The result can be readily extended to an infinite function class by using the covering number argument. For example, via the covering number argument, one can simply replace the $|\Pi|$ by the corresponding $\beta$-covering number and add a linear term of $\beta n$. As a direct application of the above lemma, the MLE estimator that minimizes $\widehat{L}^{(n)}(\theta)$ has an estimation error on the order of $c(\varepsilon)^2 \log(|\Theta|)$, with $c(\varepsilon)^2$ being the *optimal* privacy cost.

**Implications.** This lemma has several important and concrete implications. First, it helps to clarify our current understanding of MLE with log loss under privacy. Specifically, some previous works have shown that one needs to construct a new de-biased loss to work with local privacy, which turns out to be unnecessary. For instance, in the context of reward model learning with private preference data, Chowdhury et al. (2024b) attempted to work with the MLE estimator directly, but only obtained a suboptimal rate. This motivates their new de-biased loss design, which was later adopted in the alignment setting as well (Zhou et al., 2025a). However, with our above new result, we can show that the MLE estimator under local privacy can also yield an optimal rate for the reward model, by a simple application of the mean-value theorem. Further, via the reduction framework in Zhou et al. (2025a), our result can be directly leveraged in the offline alignment problem with linear function approximation, see more discussions on these in Appendix F. Second, in Section 4, we will show that the above lemma can also be utilized to design new algorithms with log loss for both offline and online alignment, even with a general function class. Finally, given the wide application of log loss in decision-making problems (Foster & Rakhlin, 2023; Foster et al., 2024), our result could find wide applications beyond alignment.

### 3.2. Square Loss under CTL and LTC

In this subsection, our main result is the following lemma, which gives the uniform convergence with square loss under *both local privacy and adversary corruption*. This can be viewed as a variant of the standard generalization bound of least squares (e.g., Agarwal et al. (2019)).

**Lemma 3.3.** *Suppose that* $\mathcal{H} \subseteq (\mathcal{X} \to [-1, 1])$ *is a given finite function class. Consider the clean data* $(x^{(1)}, y^{(1)}), \ldots, (x^{(T)}, y^{(T)})$ *be a sequence of random variables with* $x^{(t)} \in \mathcal{X}$, $y^{(t)} \in \{-1, 1\}$ *that are adapted to a filtration* $(\mathcal{F}^{(t)})_{t=1}^T$ *such that there exists* $h^\star \in \mathcal{H}$ *with* $h^\star(x^{(t)}) = \mathbb{E}[y^{(t)}|\mathcal{F}^{(t-1)}, x^{(t)}]$ *almost surely. Then, under* CTL *and* LTC *with parameters* $\alpha, \varepsilon$, *the observed data sequence is* $(x^{(1)}, z^{(1)}), \ldots, (x^{(T)}, z^{(T)})$ *such that the clean label* $y^{(t)}$ *is turned to* $z^{(t)}$ *(cf. Def. 2.5). Define*

$$\mathcal{E}^{(n)}(h) := \sum_{t=1}^n \mathbb{E}\left[ \left( h(x^{(t)}) - h^\star(x^{(t)}) \right)^2 \mid \mathcal{F}^{(t-1)} \right].$$

*Then, with probability at least* $1 - \delta$, *it holds that for all* $n \in [T]$ *and for all* $h \in \mathcal{H}$, *under* CTL *and* LTC

$$\mathcal{E}_{\mathsf{CTL}}^{(n)}(h) \lesssim \widehat{L}_{\mathsf{sq}}^{(n)}(h) - \widehat{L}_{\mathsf{sq}}^{(n)}(h^\star) + c(\varepsilon)^2 \log(|\mathcal{H}|\delta^{-1}) + n\alpha^2,$$
$$\mathcal{E}_{\mathsf{LTC}}^{(n)}(h) \lesssim \widehat{L}_{\mathsf{sq}}^{(n)}(h) - \widehat{L}_{\mathsf{sq}}^{(n)}(h^\star) + c(\varepsilon)^2 \log(|\mathcal{H}|\delta^{-1}) + nc(\varepsilon)^2\alpha^2,$$

*where* $\widehat{L}_{\mathsf{sq}}^{(n)}(h') := \sum_{t=1}^n (h'(x^{(t)}) - c(\varepsilon)z^{(t)})^2$, *for any* $h' \in \mathcal{H}$.

*Remark* 3.4. As before, we focus on the finite function class case. The result can be readily extended to an infinite function class by using the covering number argument. For instance, for the linear model in $d$ dimension, one can replace the $\log(|\mathcal{H}|)$ term by $\widetilde{O}(d)$. As a direct application of the above lemma, one can obtain that the estimation error for the least-square estimator (which minimizes $\widehat{L}_{\mathsf{sq}}^{(n)}(\theta)$) has an additional $c(\varepsilon)^2$ in the privacy case, as well as an $n\alpha^2$ bias for CTL while a larger bias term of $nc(\varepsilon)^2\alpha^2$ for LTC.

*Remark* 3.5. All these additional factors are, in fact, *optimal* by leveraging the results for mean estimations in CTL and LTC (Zhou & Zhang, 2024), with the reduction from mean estimation to regression. Specifically, the lower bound for the regression problem in Lemma 3.3 can be understood by viewing it as a special case of one-dimensional mean estimation (setting $d = 1$ and the feature $x = 1$). According to the results of Zhou & Zhang (2024), under CTL and LTC settings, the lower bounds for mean estimation error are $\Omega\big(c(\varepsilon) \cdot 1/\sqrt{n} + \alpha\big)$ and $\Omega\big(c(\varepsilon) \cdot 1/\sqrt{n} + c(\varepsilon)\alpha\big)$, respectively. Through squaring and a simple transformation with respect to $n$, this directly proves the optimality of the least squares estimator in Lemma 3.3.

**Improvement over the prior art.** The above lemma improves the state-of-the-art (Zhou et al., 2025b) in both CTL and LTC settings. In particular, for CTL, Zhou et al. (2025b) established that the corruption term is $n\alpha$, which is strictly worse than our optimal bound of $n\alpha^2$ as $\alpha \in [0, 1/2)$. For LTC, Zhou et al. (2025b) showed a bound of $nc(\varepsilon)\alpha$, which is again worse than ours for the meaningful range of $c(\varepsilon)\alpha \lesssim 1$. In Section 5, we will apply the above lemma to both offline and online alignment to arrive at state-of-the-art results.

## 4. Private Alignment: Log Loss

This section presents two log-loss–based algorithms with theoretical guarantees for alignment under private labels: one for the offline setting and the other for the online setting. Each algorithm is obtained through a simple yet principled modification of its non-private counterpart, and their guarantees are established using our earlier result, i.e., Lemma 3.1.

### 4.1. Offline Setting

In this subsection, we focus on the offline setting and present $\mathrm{Priv}\chi\mathrm{PO}$ (Algorithm 1), which is the private version of $\chi\mathrm{PO}$ proposed in Huang et al. (2024).

$\mathrm{Priv}\chi\mathrm{PO}$ takes as input a preference dataset $\widetilde{\mathcal{D}}_{\mathsf{pref}}$ with privatized labels. It first ranks the two responses using the private label and then computes the reparameterization function $h_{\chi\mathrm{PO}}^{(i)}(\pi)$ using the function $\phi(u)$, which has an additional linear term $u$ compared to the standard $\log(u)$

---

**Algorithm 1** $\mathrm{Priv}\chi\mathrm{PO}$

1: **Input:** Locally private preference dataset $\widetilde{\mathcal{D}}_{\mathsf{pref}} = \{\tau_{-1}^{(i)}, \tau_1^{(i)}, \widetilde{y}^{(i)}\}_{i=1}^n$, privacy parameter $\varepsilon > 0$, regularization coefficient $\beta > 0$, reference policy $\pi_{\mathsf{ref}}$

2: Define

$$\tau_+^{(i)} := \tau_{\widetilde{y}^{(i)}}^{(i)} \text{ and } \tau_-^{(i)} := \tau_{-\widetilde{y}^{(i)}}^{(i)}$$

$$\phi(u) := u + \log u$$

$$h_{\chi\mathrm{PO}}^{(i)}(\pi) := \beta\phi\left(\frac{\pi(\tau_+^{(i)})}{\pi_{\mathsf{ref}}(\tau_+^{(i)})}\right) - \beta\phi\left(\frac{\pi(\tau_-^{(i)})}{\pi_{\mathsf{ref}}(\tau_-^{(i)})}\right)$$

$$P_{\chi\mathrm{PO}}^{(i)}(\pi) := \sigma\left(\mathrm{clip}_{2R_{\max}}\left[h_{\chi\mathrm{PO}}^{(i)}(\pi)\right]\right)$$

3: Optimize the following objective:

$$\widehat{\pi} \leftarrow \underset{\pi \in \Pi}{\arg\max} \sum_{i \in [n]} \log\left[(2\sigma(\varepsilon)-1)P_{\chi\mathrm{PO}}^{(i)}(\pi)+(1 - \sigma(\varepsilon))\right],$$

where $\sigma(\varepsilon) := \frac{e^\varepsilon}{e^\varepsilon+1}$

4: **Output:** $\widehat{\pi}$

---

term in $\mathrm{DPO}$ (Rafailov et al., 2023). This additional term arises due to the use of $\chi^2$-divergence (in addition to standard KL-divergence) when regularizing the final policy with respect to $\pi_{\mathsf{ref}}$ in RLHF. Next, the preference probability $P_{\chi\mathrm{PO}}^{(i)}(\pi)$ is computed based on the BT-model (cf. (1)), with the truncated $h_{\chi\mathrm{PO}}^{(i)}(\pi) \in [-2R_{\max}, 2R_{\max}]$ being the reward difference. Here, the truncation is adopted to slightly improve the final bound, as in Huang et al. (2024). Finally, the policy is found by the private MLE, inspired by $\widehat{L}^{(n)}$ in Lemma 3.1, which reduces to the standard log loss when $\varepsilon = \infty$, i.e., non-private case. In summary, the only change compared to $\chi\mathrm{PO}$ is the loss objective.

In the following, we will present the sample complexity guarantee for $\mathrm{Priv}\chi\mathrm{PO}$. To this end, we first make several standard assumptions as in the non-private case (Huang et al., 2024). The first one is the *realizability* assumption, which states that the policy class $\Pi$ contains the optimal policy under regularization.

**Assumption 4.1** (Policy realizability). Fix $\beta > 0$. The policy class $\Pi$ satisfies $\pi_\beta^\star \in \Pi$, where $\pi_\beta^\star$ is the optimal policy of the following mixed $\chi^2$-regularized and KL-regularized objective:

$$J_\beta^{\chi^{\mathsf{mix}}}(\pi) := \mathbb{E}_\pi[r^\star(\tau)] - \beta \cdot [D_{\chi^2}(\pi\|\pi_{\mathsf{ref}})+D_{\mathsf{KL}}(\pi\|\pi_{\mathsf{ref}})].$$

The next assumption asserts that the *implicit reward difference* under any policy in $\Pi$ is upper bounded by some constant.

**Assumption 4.2** (Bounded implicit reward difference). For a parameter $V_{\max} \geq R_{\max}$, it holds that for all $\pi \in \Pi$,

trajectories $\tau, \tau'$ from the same state,

$$\left| \beta \phi \left( \frac{\pi(\tau)}{\pi_{\text{ref}}(\tau)} \right) - \beta \phi \left( \frac{\pi(\tau')}{\pi_{\text{ref}}(\tau')} \right) \right| \leq V_{\max}.$$

Finally, as is typical in offline RL, we need some notion of *concentrability* to measure the quality of the offline data. Following Huang et al. (2024), our main result relies on the following one.

**Definition 4.3** ($L_1$-Concentrability). For a target policy $\pi$, the single-policy $L_1$-concentrability coefficient is given by

$$\mathcal{C}^\pi := \mathbb{E}_\pi \left[ \frac{\pi(\tau)}{\pi_{\text{ref}}(\tau)} \right],$$

where we recall that $\pi(\tau)$ is a shorthand of $\pi(\tau|s)$ and $\mathbb{E}_\pi[\cdot] := \mathbb{E}_{s\sim\rho, \tau\sim\pi(\cdot|s)}[\cdot]$.

Now, the main sample complexity bound for $\text{Priv}\chi\text{PO}$ is as follows, with proof in Appendix D.1.

**Theorem 4.4.** *For any given comparator policy $\pi^\star$, there exists a proper choice of $\beta > 0$ such that when Assumptions 4.1 and 4.2 hold, with probability at least $1 - \delta$, the output of Algorithm 1 satisfies the following suboptimality gap:*

$$J(\pi^\star) - J(\widehat{\pi}) \lesssim \kappa(\pi^\star) \left( c(\varepsilon) \sqrt{\frac{\log(|\Pi|/\delta)}{n}} \right)$$

*where $c(\varepsilon) = \frac{e^\varepsilon + 1}{e^\varepsilon - 1}$ and $\kappa(\pi^\star) := e^{2R_{\max}} \cdot \frac{V_{\max}}{R_{\max}} \sqrt{\mathcal{C}^{\pi^\star}}$ is the single-policy concentrability related term.*

This theorem shows that the cost due to local label privacy is a multiplicative factor $c(\varepsilon)$, which is indeed optimal up to $\kappa(\pi^\star)$ factor. We highlight that this is the first result showing that a standard log-loss MLE-type algorithm can yield a near-optimal rate under privacy, without the design of any de-biased loss. The dependence of single-policy concentrability (rather than the larger all-policy concentrability) is the key benefit of using the additional $\chi^2$-regularization.

### 4.2. Online Setting

In this subsection, we turn to the online alignment setting. Our main algorithm is $\text{PrivXPO}$ (Algorithm 2), which is a simple modification to the non-private counterpart $\text{XPO}$ in Xie et al. (2024).

To better present the online privacy protection process, we divide the algorithm into two parts: the user side and the learner side in Algorithm 2. For each step $t \in [T]$, user $t$ submits a question $s_1^{(t)}$ and receives two responses[1]

---

[1]This two-response interaction indeed often pops up in the ChatGPT app or website.

$\tau^{(t)} \sim \pi^{(t)} \mid s_1^{(t)}$, and $\widetilde{\tau}^{(t)} \sim \pi_{\text{ref}} \mid s_1^{(t)}$. Then, user $t$ generates the true preference label and the private one, respectively. On the learner side, after receiving the $t$-th new data, it ranks the responses, updates the online dataset, and computes the probability $P_{\text{XPO}}^{(i)}$. Here, the reparameterization function is just the standard $\text{DPO}$-type, i.e., only the log function in $h_{\text{XPO}}^{(i)}$. This probability is then used to construct the empirical loss $\widehat{L}_{\text{XPO}}^{(t)}$, similar to the offline setting, which is again inspired by Lemma 3.1. Finally, the learner updates the policy by minimizing a composite loss function, which adds an additional term $\gamma \sum_{i\in[t]} \log \pi(\widetilde{\tau}^{(i)})$ to solicit active exploration via global optimism, which is the key component in $\text{XPO}$. As before, the only fundamental change compared with $\text{XPO}$ is the loss objective.

In the following, we turn to the theoretical guarantee of $\text{PrivXPO}$. To this end, we will make the same set of statistical assumptions as in $\text{XPO}$ (Xie et al., 2024).

**Assumption 4.5** (Policy realizability). Fix $\beta > 0$. The policy class $\Pi$ satisfies $\pi_\beta^\star \in \Pi$, where $\pi_\beta^\star$ is the optimal policy of the KL-regularized objective as in (2).

*Remark* 4.6 (Approximate realizability). The realizability assumption is used only to avoid carrying approximation terms. If the target model is not exactly contained in the considered class, the same proof yields an additional approximation term. For example, if

$$\Delta_{\text{app}} := \inf_{\pi \in \Pi} \mathcal{E}(\pi, \pi^\star)$$

denotes the best-in-class approximation error, then the right-hand side of the corresponding suboptimality bound is enlarged by an additive term depending on $\Delta_{\text{app}}$. Thus, the guarantees degrade gracefully under model misspecification.

We also need the following boundedness assumption.

**Assumption 4.7** (Bounded density ratios). For a parameter $V_{\max}$, it holds that for all $\pi \in \Pi$, any trajectory $\tau$,

$$\left| \beta \log \left( \frac{\pi(\tau)}{\pi_{\text{ref}}(\tau)} \right) \right| \leq V_{\max}.$$

Note that this is slightly stronger than the one in Assumption 4.2, which only requires the difference to be bounded.

Finally, we require a notion to quantify the exploration difficulty as in online RL, which is necessary for RL with general function approximations. We follow $\text{XPO}$ to use a condition known as *coverability* (Xie et al., 2022), which is formally defined below.

**Definition 4.8** (Coverability). The trajectory-level coverability coefficient is given by

$$C_{\text{cov}}(\Pi) := \inf_{\mu \in \Delta(\mathcal{T})} \sup_{\tau \in \mathcal{T}} \sup_{\pi \in \Pi} \frac{d^\pi(\tau)}{\mu(\tau)},$$

**Algorithm 2** `PrivXPO`

---

1: **Input:** Privacy parameter $\varepsilon > 0$, number of iterations $T$, regularization coefficient $\beta > 0$, optimism parameter $\gamma > 0$
2: Initialize $\pi^{(1)} \leftarrow \pi_{\mathsf{ref}}, \widetilde{\mathcal{D}}_{\mathsf{pref}}^{(0)} \leftarrow \varnothing$
3: **for** iteration $t = 1, 2, \ldots, T$ **do**
4:    /* User Side */
5:    Generate prompt/question: $s_1^{(t)} \sim \rho$,
6:    Receive two responses: $\tau^{(t)} \sim \pi^{(t)} \mid s_1^{(t)}$, and $\widetilde{\tau}^{(t)} \sim \pi_{\mathsf{ref}} \mid s_1^{(t)}$
7:    Generate true label: $y^{(t)} = 1$ w.p. $\mathcal{P}^\star(\tau^{(t)} \succ \widetilde{\tau}^{(t)} \mid s)$ as in (1); otherwise $y^{(t)} = -1$
8:    Generate private label $\widetilde{y}^{(t)}$ via RR as in (3)
9:    /* Learner Side */
10:    Receive new data $(\tau^{(t)}, \widetilde{\tau}^{(t)}, \widetilde{y}^{(t)})$ from the user
11:    Rank two responses: $(\tau^{(t)}, \widetilde{\tau}^{(t)})$ as $(\tau_+^{(t)}, \tau_-^{(t)})$ if $\widetilde{y}^{(t)} = 1$; otherwise, reserve the order
12:    Update preference data: $\widetilde{\mathcal{D}}_{\mathsf{pref}}^{(t)} = \widetilde{\mathcal{D}}_{\mathsf{pref}}^{(t-1)} \cup \{(\tau_+^{(t)}, \tau_-^{(t)})\}$
13:    Define

$$h_{\mathrm{XPO}}^{(i)}(\pi) := \beta \log\left(\frac{\pi(\tau_+^{(i)})}{\pi_{\mathsf{ref}}(\tau_+^{(i)})}\right) - \beta \log\left(\frac{\pi(\tau_-^{(i)})}{\pi_{\mathsf{ref}}(\tau_-^{(i)})}\right)$$

$$P_{\mathrm{XPO}}^{(i)}(\pi) := \sigma\left(h_{\mathrm{XPO}}^{(i)}\right)$$

$$\widehat{L}_{\mathrm{XPO}}^{(t)}(\pi) := \sum_{i \in [t]} \log\left[(2\sigma(\varepsilon) - 1)P_{\mathrm{XPO}}^{(i)} + (1 - \sigma(\varepsilon))\right]$$

14:    Optimize with global optimism:

$$\pi^{(t+1)} \leftarrow \operatorname*{argmin}_{\pi \in \Pi}\left\{\gamma \sum_{i \in [t]} \log \pi(\widetilde{\tau}^{(i)}) - c(\varepsilon)^2 \widehat{L}_{\mathrm{XPO}}^{(t)}(\pi)\right\}.$$

15: **end for**
16: **Output:** $\widehat{\pi} = \operatorname{argmax}_{\pi \in \{\pi^{(1)}, \ldots, \pi^{(T+1)}\}} J_\beta(\pi)$

---

where $\mathcal{T}$ is the trajectory space and $d^\pi(\tau)$ is the probability of observing a trajectory $\tau$ under policy $\pi$.

This coverability notion for online RL is closely related to concentrability in offline RL in that coverability can be viewed as the best concentrability in a certain case (due to the inf operator in the definition), see Xie et al. (2022) for more discussions.

We are now ready to state the following sample complexity guarantee of `PrivXPO`, with its proof given by Appendix D.2.

**Theorem 4.9.** *Suppose that Assumptions 4.5 and 4.7 hold. For any $\beta > 0$ and $T \in \mathbb{N}$, there exists a proper choice of $\gamma$ such that Algorithm 2 ensures that with probability at least $1 - \delta$,*

$$J_\beta(\pi_\beta^\star) - J_\beta(\widehat{\pi}) \lesssim \kappa_{\mathsf{cov}}(\Pi)\left(c(\varepsilon)\sqrt{\frac{\log(|\Pi|T/\delta)}{T}}\right),$$

*where $\kappa_{\mathsf{cov}}(\Pi) := (V_{\max} + R_{\max})e^{2R_{\max}}\log(T)\sqrt{C_{\mathsf{cov}}(\Pi)}$ is the coverability-related term.*

The above theorem gives the first sample complexity result for *online* alignment with label privacy. When compared with the offline counterpart in Theorem 4.4, we can see that, roughly, the online guarantee above replaces the concentrability-related term by the coverability-related term, which indicates the benefit of active online exploration. As before, the additional $c(\varepsilon)$ factor is the cost due to privacy protection, compared to the non-private XPO in Xie et al. (2024).

*Remark* 4.10 (Log-linear policy classes). The finite-class dependence in Theorem 4.9 can be replaced by standard covering-number complexity. For example, suppose $\Pi$ is a log-linear policy class parameterized by a $d$-dimensional vector with bounded norm and uniformly bounded features. Then, by replacing the finite-class union bound with a standard covering-number argument, the $\log|\Pi|$ term in Theorem 4.9 can be replaced, up to logarithmic factors, by an $\widetilde{O}(d)$ complexity term. Algorithm 2 satisfies, with probability at least $1 - \delta$,

$$J_\beta(\pi_\beta^\star) - J_\beta(\widehat{\pi}) \lesssim \kappa_{\mathsf{cov}}(\Pi)c(\varepsilon)\sqrt{\frac{\widetilde{O}(d) + \log(T/\delta)}{T}}.$$

*Remark* 4.11. We remark that the above result holds for any regularization parameter $\beta > 0$ as in the non-private case (Xie et al., 2024). However, for large $\beta > 0$, one can leverage the additional property of local strong convexity of the regularized objective to derive improved bounds, see Appendix F for detailed results and discussions.

## 5. Private and Robust Alignment: Square Loss

In this section, we turn to study alignment under both privacy protection and adversarial corruption by considering CTL and LTC settings. The only major difference compared to the last section is a square loss in place of the log loss.

A natural question is *why we do not include the private and robust alignment results for log-loss*. The key reason is that the log loss is unbounded, while the square loss is bounded, which enables us to control the effect of Huber corruption easily. That being said, whether one can establish some guarantees for log loss under Huber corruption is an interesting question, given its wide use in practice. This represents

**Algorithm 3** Square$\chi$PO (Zhou et al., 2025b)

1: **Input:** Locally private and corrupted preference dataset $\bar{\mathcal{D}}_{\mathsf{pref}} = \{\tau_{-1}^{(i)}, \tau_1^{(i)}, z^{(i)})\}_{i=1}^n$ under CTL or LTC, privacy parameter $\varepsilon > 0$, regularization coefficient $\beta > 0$, reference policy $\pi_{\mathsf{ref}}$

2: Define

$$\phi(u) := u + \log u$$

$$\widehat{h}_{\chi\mathsf{PO}}^{(i)}(\pi) := \beta\phi\left(\frac{\pi(\tau_1^{(i)})}{\pi_{\mathsf{ref}}(\tau_1^{(i)})}\right) - \beta\phi\left(\frac{\pi(\tau_{-1}^{(i)})}{\pi_{\mathsf{ref}}(\tau_{-1}^{(i)})}\right)$$

$$\widehat{P}_{\chi\mathsf{PO}}^{(i)}(\pi) := \sigma\left(\mathrm{clip}_{2R_{\max}}\left[\widehat{h}_{\chi\mathsf{PO}}^{(i)}(\pi)\right]\right)$$

3: Optimize the following *square loss* objective:

$$\widehat{\pi} \leftarrow \underset{\pi \in \Pi}{\mathrm{argmin}} \sum_{i \in [n]} \left[2\widehat{P}_{\chi\mathsf{PO}}^{(i)}(\pi) - 1 - c(\varepsilon)z^{(i)}\right]^2$$

where $\sigma(\varepsilon) := \frac{e^\varepsilon}{e^\varepsilon + 1}$

4: **Output:** $\widehat{\pi}$

---

a fundamental technical challenge that we identify as an open direction.

### 5.1. Offline Setting

In the offline setting, we will show that the existing algorithm (i.e., Square$\chi$PO in Zhou et al. (2025b)) actually enjoys an improved sample complexity bound, as an application of Lemma 3.3.

We recap Square$\chi$PO in Algorithm 3 using our notations. It essentially replaces the log loss in Priv$\chi$PO (and $\chi$PO) with a square loss. Note that to match the specific form of square loss in Lemma 3.3, we do not rank the responses using $+/-$ here. As mentioned before, one key advantage of the square loss in Algorithm 3 is that it is bounded, which helps us to handle the corruption easily.

*Remark* 5.1 (Adaptivity). We remark that Square$\chi$PO is adaptive to the actual setting (CTL or LTC) in that it does not require the knowledge of the specific setting in advance, as well as the corruption level $\alpha$.

Next, as a direct application of Lemma 3.3, we will establish an improved bound for Square$\chi$PO compared to the one in (Zhou et al., 2025b). In particular, under the same assumptions as before, we have the following sample complexity guarantee. See Appendix E.1 for the proof.

**Theorem 5.2.** *For any given comparator policy $\pi^\star$, there exists a proper choice of $\beta > 0$ such that when Assumptions 4.1 and 4.2 hold, with probability at least $1 - \delta$, the output of Algorithm 3 satisfies the following suboptimality*

gaps for CTL and LTC:

$$J(\pi^\star) - J(\widehat{\pi}_{\mathsf{CTL}}) \lesssim \kappa(\pi^\star)\left(c(\varepsilon)\sqrt{\frac{\log(|\Pi|/\delta)}{n}} + \alpha\right)$$

$$J(\pi^\star) - J(\widehat{\pi}_{\mathsf{LTC}}) \lesssim \kappa(\pi^\star)\left(c(\varepsilon)\sqrt{\frac{\log(|\Pi|/\delta)}{n}} + c(\varepsilon)\alpha\right)$$

*where $c(\varepsilon)$ and $\kappa(\pi^\star)$ are the same as before.*

The above theorem gives better bounds for both CTL and LTC, compared with Zhou et al. (2025b). For CTL, it improves from $\sqrt{\alpha}$ to $\alpha$ and for LTC, it improves from $\sqrt{c(\varepsilon)\alpha}$ to $c(\varepsilon)\alpha$. An astute reader may already observe that this essentially comes from the improvement in Lemma 3.3 over the previous result.

*Remark* 5.3. Up to $\kappa(\pi^\star)$ factor, the above bounds are in fact tight by existing lower bounds for simpler problems in CTL and LTC, see Zhou & Zhang (2024). The offline alignment problem is at least as hard as the offline multi-armed bandit (MAB) problem, since the former only obtains preference information while the latter can obtain complete reward information. Therefore, we can directly leverage the lower bounds for offline MAB under CTL and LTC settings from Zhou & Zhang (2024) (since preference learning involves two samples, the lower bound needs to be multiplied by a constant factor of 2). This demonstrates that the rate achieved by Theorem 5.2 is unimprovable in terms of the relevant parameters.

### 5.2. Online Setting

In the online setting with both privacy and corruption, we again adopt a square loss but in XPO to propose SquareXPO (Algorithm 4). This new algorithm is the first one with theoretical guarantees that can handle both privacy and corruption in the *online* setting.

SquareXPO shares the same flow as in PrivXPO in the last section. We still divide it into two parts: the user side and the learner side. For each step $t \in [T]$, on the user side, the private and corrupted preference label is generated following CTL or LTC. On the learner's side, the optimization objective changes from a log loss to a square loss. Finally, the policy is updated with the global optimism for exploration.

We have the following sample complexity bound for SquareXPO, with proof in Appendix E.2.

**Theorem 5.4.** *Suppose that Assumptions 4.5 and 4.7 hold. For any $\beta > 0$ and $T \in \mathbb{N}$, there exists a proper choice of $\gamma$ such that Algorithm 4 ensures that with probability at least*

---

**Algorithm 4** `SquareXPO`

---

1: **Input:** Privacy parameter $\varepsilon > 0$, number of iterations $T$, regularization coefficient $\beta > 0$, optimism parameter $\gamma > 0$
2: Initialize $\pi^{(1)} \leftarrow \pi_{\text{ref}}, \bar{\mathcal{D}}_{\text{pref}}^{(0)} \leftarrow \varnothing$
3: **for** iteration $t = 1, 2, \ldots, T$ **do**
4:    /* User Side */
5:    Generate prompt/question: $s_1^{(t)} \sim \rho$,
6:    Receive two responses: $\tau^{(t)} \sim \pi^{(t)} \mid s_1^{(t)}$, and $\widetilde{\tau}^{(t)} \sim \pi_{\text{ref}} \mid s_1^{(t)}$
7:    Generate true label: $y^{(t)} = 1$ w.p. $\mathcal{P}^\star(\tau^{(t)} \succ \widetilde{\tau}^{(t)} \mid s)$ as in (1); otherwise $y^{(t)} = -1$
8:    *Corrupt label if under* CTL
9:    Generate private label $\widetilde{y}^{(t)}$ via RR as in (3)
10:   *Corrupt label if under* LTC
11:   /* Learner Side */
12:   Receive new data $(\tau^{(t)}, \widetilde{\tau}^{(t)}, z^{(t)})$ from the user
13:   Update preference data: $\bar{\mathcal{D}}_{\text{pref}}^{(t)} = \bar{\mathcal{D}}_{\text{pref}}^{(t-1)} \cup \{(\tau^{(t)}, \widetilde{\tau}^{(t)}, z^{(t)}\}$
14:   Define

$$\widehat{h}_{\text{XPO}}^{(i)}(\pi) := \beta \log\left(\frac{\pi(\tau^{(i)})}{\pi_{\text{ref}}(\tau^{(i)})}\right) - \beta \log\left(\frac{\pi(\widetilde{\tau}^{(i)})}{\pi_{\text{ref}}(\widetilde{\tau}^{(i)})}\right)$$

$$\widehat{P}_{\text{XPO}}^{(i)}(\pi) := \sigma\left(\widehat{h}_{\text{XPO}}^{(i)}\right)$$

$$\widehat{L}_{\text{sqXPO}}^{(t)}(\pi) := \sum_{i \in [t]} \left[2\widehat{P}_{\text{XPO}}^{(i)}(\pi) - 1 - c(\varepsilon)z^{(i)}\right]^2$$

15:   Optimize with global optimism:

$$\pi^{(t+1)} \leftarrow \underset{\pi \in \Pi}{\arg\min}\left\{\gamma \sum_{i \in [n]} \log \pi(\widetilde{\tau}^{(i)}) - \widehat{L}_{\text{sqXPO}}^{(t)}(\pi)\right\}$$

16: **end for**
17: **Output:** $\widehat{\pi} = \arg\max_{\pi \in \{\pi^{(1)}, \ldots, \pi^{(T+1)}\}} J_\beta(\pi)$

---

$1 - \delta$ *under* CTL *and* LTC,

$$J_\beta(\pi_\beta^\star) - J_\beta(\widehat{\pi}_{\text{CTL}}) \lesssim \kappa_{\text{cov}}(\Pi)\left(c(\varepsilon)\sqrt{\frac{\log(|\Pi|T/\delta)}{T}} + \alpha\right)$$

$$J_\beta(\pi_\beta^\star) - J_\beta(\widehat{\pi}_{\text{LTC}}) \lesssim \kappa_{\text{cov}}(\Pi)\left(c(\varepsilon)\sqrt{\frac{\log(|\Pi|T/\delta)}{T}} + c(\varepsilon)\alpha\right),$$

*where $\kappa_{\text{cov}}(\Pi)$ is the same as before.*

*Remark* 5.5. The martingale-based proof of the online guarantee continues to hold under a history-dependent extension in which, at round $t$, the bad distribution may depend on the past filtration $\mathcal{F}^{(t-1)}$. Thus, the stated online result extends beyond oblivious corruptions to this form of history-dependent corruption.

This theorem shows that similar sample complexity difference between CTL and LTC maintains in the online setting by our `SquareXPO`. This separation is in fact optimal (up to $\kappa_{\text{cov}}(\Pi)$) by reusing existing lower bound for simpler online MAB problem (Zhou & Zhang, 2024). Once again, this near-optimal result is obtained via our Lemma 3.3.

*Remark* 5.6. There is a subtle difference in the sub-optimality/regret analysis for online alignment: the lower bound in Zhou & Zhang (2024) is for the unregularized objective, while Theorem 5.4 addresses the regularized objective. For small $\beta$ regime, the difference between the two objectives is very small, and the result of Theorem 5.4 is constrained by the lower bound for online MAB, which cannot be further improved. For large $\beta$ regime, faster rates can be achieved by exploiting the local strong convexity of the KL divergence (see Proposition F.1), and we believe this result also achieves optimality.

## 6. Conclusion

We presented a theoretical analysis for private and robust alignment of language models, analyzing both the offline and online settings. Our results resolve several important questions in the literature: we showed that the standard log-loss MLE algorithm can indeed achieve near-optimal rates under privacy constraints, we improved guarantees in both the corruption-only and corruption–privacy settings by leveraging uniform convergence under square loss, and we extended these results to the online regime with active exploration.

Looking forward, several promising directions remain. First, an important next step is to extend our empirical evaluation to broader model scales and additional privacy-and-corruption regimes. Second, our results are established under the classical Huber corruption model; developing theoretical guarantees under stronger *adaptive* corruption models remains an open challenge for general function approximations (Zhou et al., 2025a). Finally, our algorithms could be extended to hybrid settings that combine offline datasets with online preference feedback to enjoy additional benefits.

## Acknowledgement

XZ would like to thank Francesco Orabona on discussion about the MLE loss. This work is supported by NSF grants under CAREER-2441519, CNS-2312835, and CNS-2153220.

## Impact Statement

This paper presents work whose goal is to advance the field of theoretical Reinforcement Learning from human feed-

back. There are many potential societal consequences of our work, none of which we feel must be specifically highlighted here.

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

# A. Detailed Related Work

In this section, we review several different topics related to our paper.

**Differentially private bandits and RL.** The theoretical foundations of private alignment are closely connected to a broader literature on reinforcement learning and bandits under differential privacy constraints. Early work primarily focuses on multi-armed bandits (MABs), establishing near-optimal regret bounds under local or central differential privacy (Mishra & Thakurta, 2015; Sajed & Sheffet, 2019; Azize & Basu, 2022; Ren et al., 2020; Wu et al., 2023; Chowdhury & Zhou, 2022b). These results were subsequently extended to contextual bandits (Shariff & Sheffet, 2018; He et al., 2022; Chowdhury & Zhou, 2022c; Zhou & Chowdhury, 2023; Chen et al., 2025b), where perturbing rewards, gradients, or sufficient statistics typically enforce privacy. Beyond bandits, a growing body of work studies differentially private reinforcement learning in episodic MDPs (Vietri et al., 2020; Luyo et al., 2021; Chowdhury & Zhou, 2022a; Qiao & Wang, 2023). Among these, only a small number of papers explicitly address *policy optimization* under privacy constraints. In particular, Chowdhury & Zhou (2022a) and Zhou (2022) analyze private policy optimization in exploratory settings, characterizing the cost of privacy in regret by privatizing optimistic variants of PPO or natural policy gradient under tabular and linear function approximation, respectively. More recently, He & Zhou (2025) establishes the first comprehensive set of sample complexity bounds for differentially private policy optimization.

**Standard offline and online alignment.** Theoretical analysis of alignment algorithms has become an active research area. One line of research focuses on the purely offline setting where the preference dataset is pre-collected, e.g., (Zhu et al., 2023; Xiong et al., 2023; Zhan et al., 2023a; Ye et al., 2024; Liu et al., 2024; Cen et al., 2024; Huang et al., 2024). Among them, $\chi$PO in Huang et al. (2024) will serve as our basis for the offline alignment due to its state-of-the-art result with a simple implementation, which is a simple one-line change to the DPO algorithm in Rafailov et al. (2023). One key limitation of offline alignment is that it requires the offline dataset to have a good coverage with respect to the optimal policy (or any other comparator policy). This motivates the study of online alignment, where the algorithm has access to online preference feedback, which often requires active exploration to achieve the optimal rates, e.g., (Xu et al., 2020; Zhan et al., 2023b; Ye et al., 2024; Chen et al., 2022; Cen et al., 2024; Xie et al., 2024). Among them, XPO in Xie et al. (2024) will be our basis for the online setting due to its good result achieved again via a one-line change to the DPO algorithm in Rafailov et al. (2023). We also mention that some works consider the *hybrid* setting that has both an offline dataset and online feedback (e.g., Bose et al. (2024); Song et al. (2024)), which may have further advantages.

**Private and robust alignment.** The current theoretical results in this area mainly focus on the offline setting. With a linear model assumption, Chowdhury et al. (2024b) studies the reward model learning with private randomized labels (Warner, 1965). After showing that the standard MLE-type log loss fails to achieve the optimal rate, it turns to propose a new de-biased log loss, which has been used in many follow-up works, e.g., (Chowdhury et al., 2024a; Zhou et al., 2025a). An important question here is whether the failure of the standard MLE-type log loss in Chowdhury et al. (2024b) is due to their specific analysis or to the intrinsic limitation of such a loss. Our paper shows that it is the former by giving near-optimal results for private alignment using standard MLE-type log loss. When it comes to adversary corruption of labels (e.g., Huber corruption (Huber, 1964)) and the joint-corruption-privacy case, the boundedness of the square loss (when compared with the unboundedness of log loss) becomes extremely useful. By utilizing such a property, Zhou et al. (2025b) proposed Square$\chi$PO—which modifies the log loss in $\chi$PO to a square loss—to study the interplay between corruption and privacy in offline alignment. Although it is a pioneering work, the guarantees of Square$\chi$PO turn out to be strictly suboptimal in terms of both the corruption parameter and the interplay between corruption and privacy parameters. Our paper shows that Square$\chi$PO in fact has a stronger guarantee via a new analysis. We note that a recent work (Wu et al., 2025) also establishes faster rates for the regularized objective in both offline and online settings under DP. One key difference is that no corruption is considered in their results.

# B. Additional Experiments

In this chapter, we present a small set of proof-of-concept experiments to complement our theoretical results. As with many ICML theoretical works, our primary focus is on establishing sharp theoretical guarantees rather than extensive empirical evaluations.

Nevertheless, since our analysis shows that multiple loss functions (i.e., the debiased log loss, square loss, and our MLE-type log loss) achieve the same theoretical rates in the private alignment setting, a natural question arises:

**Do these losses also exhibit comparable practical behavior when implemented under the same training pipeline?**

## B.1. Experiment Setup

We follow exactly the same dataset, model architecture, and training pipeline as in prior works on private alignment (e.g., (Zhou et al., 2025a;b)), and only modify the loss function used for preference optimization. We compare the following three approaches: *MLE-type log loss*; *Debiased log loss (Zhou et al., 2025a)* and *Square loss (Zhou et al., 2025b)* for private alignment.

**Dataset.** We utilize GPT-4o to generate a synthetic dataset, referred to as `finance_preference`, which comprises 1697 preference samples. Each sample includes a prompt related to a financial scenario and two possible responses, where "rejected" represents the high-risk option and "chosen" represents the low-risk option. This labeling can be viewed as private or sensitive information. For SFT training, we construct the `finance_sft` dataset by simply concatenating the prompt with the corresponding "chosen" response.

**SFT Training.** We begin by fine-tuning GPT2-large using the `finance_sft` dataset to obtain the SFT policy, $\pi_{\mathsf{sft}}$. For this, we directly utilize the SFT trainer from the Transformer Reinforcement Learning (TRL) library (von Werra et al., 2020), with the hyperparameters listed in Table 3.

**Evaluation.** All methods are evaluated under the same privacy budget $\varepsilon = 0.5$. Following standard practice in the alignment literature, performance is measured by the *win rate* (%) against the supervised fine-tuned policy $\pi_{\mathsf{sft}}$. For each test prompt, responses are generated by the evaluated policy and compared against the corresponding response produced by $\pi_{\mathsf{sft}}$. Pairwise comparisons are conducted using the `llama3:70b` model as an automated judge, following the evaluation protocol of Rafailov et al. (2023). The win rate is defined as the fraction of test instances for which the judge prefers the response generated by the evaluated policy over that of $\pi_{\mathsf{sft}}$.

*Table 3.* Hyperparameters used for SFT training.

| Parameter | Value |
|---|---|
| learning rate | 1e−5 |
| batch size | 8 |
| num train epochs | 3 |

All methods are evaluated under the same privacy budget $\varepsilon = 0.5$. Performance is measured by the win rate (%) against the supervised fine-tuned policy $\pi_{\mathsf{sft}}$, following standard practice in the alignment literature.

## B.2. Results

Table 4 reports the win rate (%) over $\pi_{\mathsf{sft}}$ for different loss functions.

*Table 4.* Win rate (%) over $\pi_{\mathsf{sft}}$ under different loss functions with $\varepsilon = 0.5$.

| Method | Win rate (%) |
|---|---|
| MLE-type log loss (ours) | $65.2 \pm 4.3$ |
| Debiased log loss (Zhou et al., 2025a) | $65.8 \pm 5.6$ |
| Square loss (Zhou et al., 2025b) | $64.5 \pm 1.2$ |

All reported results are averaged over 5 independent random seeds. We observe that all three approaches achieve comparable performance, which is consistent with our theoretical findings that they attain the same convergence rates in the private alignment setting. Together with our theoretical results, these experiments help clarify the practical implications of our new uniform convergence analysis and position our contributions relative to previous work.

## B.3. Additional Experiment on PKU-SafeRLHF

To complement the controlled synthetic experiment above, we further evaluate the three private-alignment objectives on a real-world safety preference dataset, `PKU-Alignment/PKU-SafeRLHF` (Ji et al., 2024). Specifically, we compare the *MLE-type log loss*, *debiased log loss (Zhou et al., 2025a)*, and *square loss (Zhou et al., 2025b)* under the same privacy budget $\varepsilon = 0.5$.

**Dataset.** We use the training split of `PKU-SafeRLHF` and retain only examples in which exactly one of the two candidate responses is marked safe. This filtering yields 10813 eligible preference pairs. We then select 1697 examples with selection seed 42, using 1442 examples for preference optimization, 85 examples for validation, and 170 held-out prompts for evaluation. The safe response is treated as "chosen" and the unsafe response as "rejected", matching the official `safer_response_id` field on all selected examples.

**SFT Training.** We fine-tune GPT2-large on the safe responses to obtain the supervised fine-tuned policy $\pi_{\mathsf{sft}}$. The SFT hyperparameters are identical to those in Table 3: learning rate $1e-5$, batch size 8, and 3 training epochs.

**Private Preference Optimization.** For each method, we run 5 independent private dataset seeds. Preference labels are privatized by randomized response under $\varepsilon = 0.5$, with flip probability $1/(1 + \exp(\varepsilon))$. In the strict private-only $\chi$PO implementation, we set $\alpha = 0$, $\beta = 0.01$, and the $\chi$PO coefficient to 0.25. All three methods use the same strict private-only $\chi$PO transformed score and the same training, generation, and evaluation protocol; only the loss function is varied. Each trained policy is then used to generate one response for each of the 170 held-out test prompts.

**Evaluation.** For consistency with the win-rate evaluation above, every generated response is paired with the corresponding response from $\pi_{\mathsf{sft}}$ using a fixed A/B randomization rule. We use `llama3:70b` as the judge model and evaluate 3 methods $\times$ 5 seeds $\times$ 170 held-out test prompts, resulting in 2550 pairwise comparisons in total. We report the win rate over all comparisons, computed as the fraction of comparisons in which the method's output is preferred by the judge.

*Table 5.* Win rate over all pairwise comparisons (%) against $\pi_{\mathsf{sft}}$ on the PKU-SafeRLHF experiment with $\varepsilon = 0.5$.

| Method | Win rate (%) |
|---|---|
| MLE-type log loss (ours) | $61.45 \pm 2.81$ |
| debiased log loss (Zhou et al., 2025a) | $61.95 \pm 2.50$ |
| square loss (Zhou et al., 2025b) | $61.64 \pm 2.25$ |

Table 5 reports the mean and standard deviation over 5 independent random seeds. The three losses achieve closely comparable win rates on this real-world safety preference dataset, consistent with the behavior observed in the synthetic finance experiment in Table 4 and with our theoretical characterization of these objectives in the private alignment setting.

## C. Missing Proofs For Section 3

### C.1. Detailed Proof of Lemma 3.1

We first state the standard non-private uniform convergence result under log loss. Similar results can be found in many existing works, e.g., (Van de Geer, 2000, Chapter 7), (Agarwal et al., 2020, Theorem 21), (Xie et al., 2024, Lemma C.6), and (Chen et al., 2025a, Proposition 8).

**Lemma C.1.** *Suppose that $\{P_\gamma(o|x)\}_{\gamma \in \Gamma} \subseteq (\mathcal{X} \rightarrow \Delta(\mathcal{O}))$ is a class of conditional densities parameterized by a finite class $\Gamma$ with $\mathcal{O}$ being discrete. Consider the data $(x^{(1)}, o^{(1)}), \ldots, (x^{(T)}, o^{(T)})$ be a sequence of random variables adapted to a filtration $(\mathcal{F}^{(t)})_{t=1}^T$ such that there exists $\gamma^\star \in \Theta$ so that $\mathbb{P}(o^t = \cdot | x^{(t)}, \mathcal{F}^{(t-1)}) = P_{\gamma^\star}(o^{(t)} = \cdot | x^{(t)})$ almost surely for $t \in [T]$. Then, with probability at least $1 - \delta$, it holds that for $n \in [T]$, for all $\gamma \in \Gamma$,*

$$\sum_{t=1}^n \mathbb{E}\left[ D_{\mathsf{H}}^2\left( P_\gamma(\cdot|x^{(t)}), P_{\gamma^\star}(\cdot|x^{(t)}) \right) \mid \mathcal{F}^{(t-1)} \right] \lesssim \widehat{L}^{(n)}(\gamma) - \widehat{L}^{(n)}(\gamma^\star) + \log(|\Gamma|\delta^{-1}),$$

*where for any given $\gamma' \in \Theta$, $\widehat{L}^{(n)}(\gamma') := \sum_{t=1}^n - \log P_{\gamma'}(o^{(t)}|x^{(t)})$.*

*Proof of Lemma 3.1.* We first aim to apply Lemma C.1 to our private data sequence $(x^{(1)}, \widetilde{y}^{(1)}), \ldots, (x^{(T)}, \widetilde{y}^{(T)})$. To this end, we can have the following mapping: for any $\theta \in \Theta$, with $\mathcal{O} = \{-1, 1\}$, we let $P_\gamma(o|x) := \widetilde{P}_\theta(\widetilde{y}|x) := \sigma(\varepsilon) \cdot P_\theta(\widetilde{y}|x) + (1 - \sigma(\varepsilon)) \cdot P_\theta(-\widetilde{y}|x)$. Given the realizability condition on the clean label $y$, with this mapping and the generation of private label $\widetilde{y}$, we have that the condition of Lemma C.1 is satisfied. Thus, we have with probability at least $1 - \delta$, for $n \in [T]$, for all $\theta \in \Theta$,

$$\sum_{t=1}^{n} \mathbb{E}\left[D_{\mathsf{H}}^2\left(\widetilde{P}_\theta(\cdot|x^{(t)}), \widetilde{P}_{\theta^\star}(\cdot|x^{(t)})\right) \mid \mathcal{F}^{(t-1)}\right] \lesssim \widehat{L}^{(n)}(\theta) - \widehat{L}^{(n)}(\theta^\star) + \log(|\Theta|\delta^{-1}),$$

where recall that $\widehat{L}^{(n)}(\theta) := \sum_{t=1}^{n} -\log \widetilde{P}_\theta(\widetilde{y}^{(t)}|x^{(t)})$.

Next, by the fact that for any two distributions $P, Q$, $\|P - Q\|_{\mathsf{TV}}^2 \leq D_{\mathsf{H}}^2(P, Q)$, we have

$$\sum_{t=1}^{n} \mathbb{E}\left[\left\|\widetilde{P}_\theta(\cdot|x^{(t)}) - \widetilde{P}_{\theta^\star}(\cdot|x^{(t)})\right\|_{\mathsf{TV}}^2 \mid \mathcal{F}^{(t-1)}\right] \lesssim \widehat{L}^{(n)}(\theta) - \widehat{L}^{(n)}(\theta^\star) + \log(|\Theta|\delta^{-1}).$$

Finally, by the binary distribution and $\widetilde{P}_\theta(\widetilde{y}|x) = (2\sigma(\varepsilon) - 1) P_\theta(\widetilde{y}|x) + (1 - \sigma(\varepsilon))$, we have

$$(2\sigma(\varepsilon) - 1)^2 \cdot \sum_{t=1}^{n} \mathbb{E}\left[\left\|P_\theta(\cdot|x^{(t)}) - P_{\theta^\star}(\cdot|x^{(t)})\right\|_{\mathsf{TV}}^2 \mid \mathcal{F}^{(t-1)}\right] \lesssim \widehat{L}^{(n)}(\theta) - \widehat{L}^{(n)}(\theta^\star) + \log(|\Theta|\delta^{-1}),$$

which gives the final result by rescaling. $\qquad\square$

## C.2. Detailed Proof of Lemma 3.3

We first state the standard non-private non-corrupted uniform convergence result under square loss. The following particular result is an adaptation based on Lemmas 6 and 7 (along with their proof) in Chen et al. (2025a). We note that one can also prove Lemma 3.3 using Freedman's inequality directly. However, we tend to believe that utilizing existing results is a more modular approach.

**Lemma C.2.** *Suppose that $\mathcal{G} \subseteq (\mathcal{X} \to [-1, 1])$ is a fixed finite function class. Consider the data $(x^{(1)}, o^{(1)}), \ldots, (x^{(T)}, o^{(T)})$ be a sequence of random variables with $x^{(t)} \in \mathcal{X}$, $o^{(t)} \in \{-C, C\}$ for some $C > 0$, which are adapted to a filtration $(\mathcal{F}^{(t)})_{t=1}^T$ such that there exists a function $F^\star : \mathcal{X} \to [-C, C]$ with $F^\star(x^{(t)}) = \mathbb{E}[o^{(t)}|\mathcal{F}^{(t-1)}, x^{(t)}]$ almost surely. Suppose that there exists a function $G^\star \in \mathcal{G}$ with $\|F^\star - G^\star\|_\infty \leq \alpha_{\mathsf{app}}$. Then, with probability at least $1 - \delta$, it holds that for all $G \in \mathcal{G}$, for all $n \in [T]$*

$$\sum_{t=1}^{n} \mathbb{E}\left[\left(G(x^{(t)}) - F^\star(x^{(t)})\right)^2 |\mathcal{F}^{(t-1)}\right] \lesssim \widehat{L}_{\mathsf{sq}}^{(n)}(G) - \widehat{L}_{\mathsf{sq}}^{(n)}(G^\star) + C^2 \log(|\mathcal{G}|\delta^{-1}) + n\alpha_{\mathsf{app}}^2,$$

*where for any $G' \in \mathcal{G}$, $\widehat{L}_{\mathsf{sq}}^{(n)}(G') := \sum_{t=1}^{n}(G'(x^{(t)}) - o^{(t)})^2$.*

*Proof of Lemma 3.3.* We start with the CTL setting. Our first step is to apply Lemma C.2 to the private and corrupted labels under CTL: $(x^{(1)}, z^{(1)}), \ldots, (x^{(T)}, z^{(T)})$. To this end, we consider the following mapping: $c(\varepsilon)z^{(t)} \to o^{(t)}$ with $C = c(\varepsilon)$ and $\mathcal{H} \to \mathcal{G}$. The remaining thing is to find $\alpha_{\mathsf{app}}$ with the corresponding $G^\star$. To proceed, under CTL, we have the following process: $y \xrightarrow{\alpha} \bar{y} \xrightarrow{\varepsilon} z$, where $\bar{y}$ is the intermediate corrupted label. Then, by the definition of randomized response and Huber corruption, we have

$$\begin{aligned}
\left|F^\star(x^{(t)}) - h^\star(x^{(t)})\right| &= \left|\mathbb{E}[c(\varepsilon)z^{(t)}|\mathcal{F}^{(t-1)}, x^{(t)}] - \mathbb{E}[y^{(t)}|\mathcal{F}^{(t-1)}, x^{(t)}]\right| \\
&= \left|\mathbb{E}[\bar{y}^{(t)}|\mathcal{F}^{(t-1)}, x^{(t)}] - \mathbb{E}[y^{(t)}|\mathcal{F}^{(t-1)}, x^{(t)}]\right| \leq 2\alpha.
\end{aligned} \tag{4}$$

Here, the equality uses the unbiasedness of the rescaled randomized-response output, and the final inequality follows from the $\alpha$-Huber corruption model together with the fact that the labels lie in $\{-1, 1\}$. Thus, with $G^\star = h^\star$, we have $\alpha_{\mathsf{app}} = 2\alpha = O(\alpha)$.

We now can apply Lemma C.2 to obtain that with probability at least $1 - \delta$, for all $h \in \mathcal{H}$, for all $n \in [T]$

$$\sum_{t=1}^{n} \mathbb{E}\left[\left(h(x^{(t)}) - F^{\star}(x^{(t)})\right)^2 | \mathcal{F}^{(t-1)}\right] \lesssim \widehat{L}_{\mathsf{sq}}^{(n)}(h) - \widehat{L}_{\mathsf{sq}}^{(n)}(h^{\star}) + c(\varepsilon)^2 \log(|\mathcal{H}|\delta^{-1}) + n\alpha^2.$$

Finally, by the fact that $(a + b)^2 \leq 2(a^2 + b^2)$ and (4), we have

$$\sum_{t=1}^{n} \mathbb{E}\left[\left(h(x^{(t)}) - h^{\star}(x^{(t)})\right)^2 | \mathcal{F}^{(t-1)}\right] \lesssim \widehat{L}_{\mathsf{sq}}^{(n)}(h) - \widehat{L}_{\mathsf{sq}}^{(n)}(h^{\star}) + c(\varepsilon)^2 \log(|\mathcal{H}|\delta^{-1}) + n\alpha^2,$$

which gives the required result under CTL.

For LTC, the proof is essentially the same, with the only key difference in $\alpha_{\mathsf{app}}$. To determine it, under LTC, we have the following process: $y \xrightarrow{\varepsilon} \widetilde{y} \xrightarrow{\alpha} z$, where $\widetilde{y}$ is the intermediate private label under RR. Then, by the definition of randomized response and Huber corruption, we have

$$\left|F^{\star}(x^{(t)}) - h^{\star}(x^{(t)})\right| = \left|\mathbb{E}[c(\varepsilon)z^{(t)}|\mathcal{F}^{(t-1)}, x^{(t)}] - \mathbb{E}[c(\varepsilon)\widetilde{y}^{(t)}|\mathcal{F}^{(t-1)}, x^{(t)}]\right|$$
$$\leq 2c(\varepsilon)\alpha. \tag{5}$$

Here, we use $\mathbb{E}[c(\varepsilon)\widetilde{y}^{(t)}|\mathcal{F}^{(t-1)}, x^{(t)}] = h^{\star}(x^{(t)})$, which follows from the unbiasedness of the rescaled randomized-response output. The final inequality follows from the $\alpha$-Huber corruption model and the fact that both $z^{(t)}$ and $\widetilde{y}^{(t)}$ lie in $\{-1, 1\}$. Thus, with $G^{\star} = h^{\star}$, we have $\alpha_{\mathsf{app}} = 2c(\varepsilon)\alpha = O(c(\varepsilon)\alpha)$.

Then, with the same argument as above, we have the final result

$$\sum_{t=1}^{n} \mathbb{E}\left[\left(h(x^{(t)}) - h^{\star}(x^{(t)})\right)^2 | \mathcal{F}^{(t-1)}\right] \lesssim \widehat{L}_{\mathsf{sq}}^{(n)}(h) - \widehat{L}_{\mathsf{sq}}^{(n)}(h^{\star}) + c(\varepsilon)^2 \log(|\mathcal{H}|\delta^{-1}) + nc(\varepsilon)^2\alpha^2.$$

$\square$

## D. Missing Proofs For Section 4

### D.1. Detailed Proof of Theorem 4.4

We first present the following meta theorem for $\mathtt{Priv}\chi\mathtt{PO}$.

**Theorem D.1.** *Suppose Assumptions 4.1 and 4.2 hold. Define $\widehat{r}(\tau) := \beta\phi\left(\frac{\widehat{\pi}(\tau)}{\pi_{\mathsf{ref}}(\tau)}\right)$ for the output $\widehat{\pi}$ of Algorithm 1. Then, we have*

$$J(\pi^{\star}) - J(\widehat{\pi}) \leq \frac{2V_{\mathsf{max}}}{R_{\mathsf{max}}}\sqrt{C^{\pi^{\star}} \cdot \mathsf{err}_{\mathsf{stat}}^2} + \beta \cdot C^{\pi^{\star}} + 2\beta^{-1} \cdot \frac{V_{\mathsf{max}}^2\mathsf{err}_{\mathsf{stat}}^2}{R_{\mathsf{max}}^2},$$

*where*

$$\mathsf{err}_{\mathsf{stat}}^2 = \mathbb{E}_{\pi_{\mathsf{ref}}, \pi_{\mathsf{ref}}}\left[\left(\mathsf{clip}_{2R_{\mathsf{max}}}[\widehat{\Delta}(\tau, \tau')] - \mathsf{clip}_{2R_{\mathsf{max}}}[\Delta^{\star}(\tau, \tau')]\right)^2\right],$$

*with $\widehat{\Delta}(\tau, \tau') := \widehat{r}(\tau) - \widehat{r}(\tau')$ and $\Delta^{\star}(\tau, \tau') := r^{\star}(\tau) - r^{\star}(\tau')$. Furthermore, by taking $\beta = \sqrt{\frac{2}{C^{\pi^{\star}}}} \cdot \frac{V_{\mathsf{max}}\mathsf{err}_{\mathsf{stat}}}{R_{\mathsf{max}}}$, we obtain*

$$J(\pi^{\star}) - J(\widehat{\pi}) \lesssim \frac{V_{\mathsf{max}}}{R_{\mathsf{max}}}\sqrt{C^{\pi^{\star}} \cdot \mathsf{err}_{\mathsf{stat}}^2}.$$

*Proof.* The above result largely follows from the proof of Theorem F.1 in Huang et al. (2024) (the latest arxiv version). The key in their proof is the translation from working with policy to working with the implicit reward $\widehat{r}$ defined above, i.e., Lemma F.2 in Huang et al. (2024). With this, one can follow the standard proof for RLHF to arrive at the above result by relying on the fact that $\mathcal{C}^{\pi} = 2D_{\chi^2}(\pi\|\pi_{\mathsf{ref}}) + 1$. Note that since our $\mathtt{Priv}\chi\mathtt{PO}$ uses the same re-parametrization function $\phi$ as in $\chi\mathtt{PO}$, the above argument via their Lemma F.2 still works. $\square$

*Proof of Theorem 4.4.* By the above meta theorem, we only need to focus on the term $\text{err}^2_{\text{stat}}$, which is the counterpart of Lemma F.1 in Huang et al. (2024). We are going to show a similar result.

First, recall that

$$\widehat{\pi} \leftarrow \underset{\pi \in \Pi}{\arg\max} \sum_{i \in [n]} \log \left[ (2\sigma(\varepsilon) - 1) P^{(i)}_{\chi\text{PO}}(\pi) + (1 - \sigma(\varepsilon)) \right],$$

where

$$P^{(i)}_{\chi\text{PO}}(\pi) = \sigma\left( \text{clip}_{2R_{\max}} \left[ \beta\, \phi\left( \frac{\pi(\tau^{(i)}_+)}{\pi_{\text{ref}}(\tau^{(i)}_+)} \right) - \beta\, \phi\left( \frac{\pi(\tau^{(i)}_-)}{\pi_{\text{ref}}(\tau^{(i)}_-)} \right) \right] \right).$$

Our first step is to bound the estimation error in probability by using Lemma 3.1, i.e.,

$$\sum_{i=1}^{n} \mathbb{E}\left[ P^{(i)}_{\chi\text{PO}}(\widehat{\pi}) - P^{(i)}_{\chi\text{PO}}(\pi^\star_\beta) \right]^2 \leq c(\varepsilon)^2 \log(|\Pi|/\delta),$$

To this end, all we need to show is that the probability of the true raw label being 1 is realized by $P^{(i)}_{\chi\text{PO}}(\pi^\star_\beta)$, since $\pi^\star_\beta \in \Pi$ (hence the empirical part is nonpositive). By the BT-model, we know that this probability is given by $\sigma(r^\star(\tau_+) - r^\star(\tau_-))$. Meanwhile, we know that $\pi^\star_\beta$ satisfies

$$r^\star(\tau) = \beta\phi\left( \frac{\pi^\star_\beta(\tau)}{\pi_{\text{ref}}(\tau)} \right) + Z_\beta(s_1),$$

where $s_1$ is the initial state. Thus, we have

$$P^{(i)}_{\chi\text{PO}}(\pi^\star_\beta) = \sigma\left( \text{clip}_{2R_{\max}} \left[ r^\star(\tau^{(i)}_+) - r^\star(\tau^{(i)}_-) \right] \right) = \sigma(r^\star(\tau^{(i)}_+) - r^\star(\tau^{(i)}_-)),$$

where the first equality holds by cancelling the common normalization term $Z_\beta(s_1)$ and the second equality holds by the boundedness of $r^\star \in [0, R_{\max}]$.

To finally get rid of $\sigma$ operator, one can use the standard mean-value theorem (cf. Lemma D.2) to have an additional factor of $O(e^{4R_{\max}})$. Hence, we have that under Algorithm 1

$$\text{err}^2_{\text{stat}} = \mathbb{E}_{\pi_{\text{ref}}, \pi_{\text{ref}}} \left[ \left( \text{clip}_{2R_{\max}}[\widehat{\Delta}(\tau, \tau')] - \text{clip}_{2R_{\max}}[\Delta^\star(\tau, \tau')] \right)^2 \right] \lesssim e^{4R_{\max}} c(\varepsilon)^2 \log(|\Pi|/\delta).$$

Plugging it into the above meta theorem yields the required result. $\square$

**Lemma D.2** (Lemma C.3 in Zhou et al. (2025b)). *For $z, z' \in [-R, R]$ and $R \geq 1$, by the mean-value theorem we have*

$$|z - z'| \leq (e^{-R} + 2 + e^R) \cdot |\sigma(z) - \sigma(z')|,$$

*where $\sigma(\cdot)$ is the sigmoid function.*

### D.2. Detailed Proof of Theorem 4.9

For the proof of Theorem 4.9, we mainly follow from Lemma 3.1, which gives the private uniform convergence result under log loss. Our proof is modular once we have the generalization error bounds. Here we first present a meta theorem, which is a simple adaptation from the proof in Xie et al. (2024) to our algorithm. We will then provide the mapping from Lemma 3.1 to Algorithm 2 here to demonstrate why we can utilize such a lemma.

**Theorem D.3** (Meta Theorem for both `PrivXPO` and `SquareXPO` under BT). *Under the BT-preference model, let Assumptions 4.5 and 4.7 hold. Suppose the following bound holds*

$$\gamma \mathbb{E}_{\pi_{\text{ref}}} \left[ \log \pi^{(t)}(\tau) - \log \pi^\star_\beta(\tau) \right] + \kappa \left[ \mathbb{E}_{s_1 \sim \rho, (\tau, \widetilde{\tau}) \sim \mu^{(t)}|s_1} (f_{\pi^{(t)}} - f_{\pi^\star_\beta})^2 \right] \lesssim \frac{\text{err}}{t-1} + \frac{\gamma}{\beta} V_{\max} \sqrt{\frac{\log(|\Pi|T\delta^{-1})}{t-1}}, \quad (6)$$

*we have*

$$J_\beta(\pi_\beta^\star) - J_\beta(\widehat{\pi}) \lesssim \frac{V_{\max}}{T} + \frac{C_{\mathsf{cov}}(\Pi)\log T}{\eta T} + \eta V_{\max}^2 + \frac{1}{T}\sum_{t=2}^{T}\left(\frac{\beta\mathsf{err}}{\gamma(t-1)} + V_{\max}\sqrt{\frac{\log(|\Pi|T\delta^{-1})}{t-1}}\right),$$

*where* $\kappa := \left(8(R_{\max} + V_{\max})e^{2R_{\max}}\right)^{-2}$, $\mu^{(t)} = \frac{1}{t-1}\sum_{i<t}\pi^{(i)}\otimes\pi_{\mathsf{ref}}$, $f_\pi(\tau,\widetilde{\tau}) = \beta\log\frac{\pi(\tau)}{\pi_{\mathsf{ref}}(\tau)} - \beta\log\frac{\pi(\widetilde{\tau})}{\pi_{\mathsf{ref}}(\widetilde{\tau})}$.

Recall the update rule in both Algorithm 2 and 4 and define $\widehat{B}^{(t)}(\pi) = \gamma\sum_{\widetilde{\tau}}\log\pi(\widetilde{\tau})$, then

$$\pi^{(t+1)} \leftarrow \operatorname*{argmin}_{\pi\in\Pi}\left\{\widehat{B}^{(t)}(\pi) - \widehat{L}^{(t)}(\pi)\right\},$$

where $\widehat{L}^{(t)}(\pi) = c(\varepsilon)^2 \cdot \widehat{L}_{\mathtt{XPO}}^{(t)}(\pi)$ for $\mathtt{PrivXPO}$, as for $\mathtt{SquareXPO}$, $\widehat{L}^{(t)}(\pi) = \widehat{L}_{\mathtt{sqXPO}}^{(t)}(\pi)$.

To satisfy Equation (6), we need the following two lemmas.

**Lemma D.4** (Adapted from Xie et al. (2024), Lemma C.6). *For any fixed $t \geq 1$, for all $\pi \in \Pi$ with probability at least $1 - \delta$, we have:*

$$\kappa \cdot \left[\mathbb{E}_{s_1\sim\rho,(\tau,\widetilde{\tau})\sim\mu^{(t)}|s_1}(f_{\pi^{(t)}} - f_{\pi_\beta^\star})^2\right] \leq \widehat{L}^{(t)}(\pi) - \widehat{L}^{(t)}(\pi_\beta^\star) + \mathsf{err}.$$

**Lemma D.5** (Adapted from Xie et al. (2024), Lemma C.7). *For any fixed $t \geq 1$, for all $\pi \in \Pi$ with probability at least $1 - \delta$, we have:*

$$\gamma(t-1)\mathbb{E}_{\pi_{\mathsf{ref}}}\left[\log\pi^{(t)}(\tau) - \log\pi_\beta^\star(\tau)\right] \leq \widehat{B}^{(t)}(\pi) - \widehat{B}^{(t)}(\pi_\beta^\star) + \frac{\gamma}{\beta}V_{\max}\sqrt{(t-1)\log(|\Pi|\delta^{-1})}.$$

Combining Lemma D.4 and Lemma D.5, taking an union bound over all time steps $t \in T$, and note that $\widehat{B}^{(t)}(\pi) - \widehat{L}^{(t)}(\pi) \leq \widehat{B}^{(t)}(\pi_\beta^\star) - \widehat{L}^{(t)}(\pi_\beta^\star)$, then we get the stated result.

*Proof of Theorem D.3.* Following Xie et al. (2024), using the regret decomposition techniques for KL-Regularized MDPs, define $\delta^{(t)}(\tau,\widetilde{\tau}) := \beta\log\frac{\pi^{(t)}(\tau)}{\pi_{\mathsf{ref}}(\tau)} - r(\tau) - \beta\log\frac{\pi^{(t)}(\widetilde{\tau})}{\pi_{\mathsf{ref}}(\widetilde{\tau})} + r(\widetilde{\tau})$, we derive

$$\begin{aligned}
J_\beta(\pi_\beta^\star) - J_\beta(\widehat{\pi}) &\leq \frac{1}{T}\sum_{t=1}^{T}J_\beta(\pi_\beta^\star) - J_\beta(\pi^{(t)})\\
&\leq \frac{6V_{\max}}{T} + \frac{1}{T}\sum_{t=2}^{T}\mathbb{E}_{\tau\sim\pi_{\mathsf{ref}}}\left[\beta\log\pi^{(t)}(\tau) - \beta\log\pi_\beta^\star(\tau)\right]\\
&\quad + \frac{1}{T}\sum_{t=2}^{T}\mathbb{E}_{s_1\sim\rho,\tau\sim\pi^{(t)}|s_1,\widetilde{\tau}\sim\pi_{\mathsf{ref}}|s_1}\left[\delta^{(t)}(\tau,\widetilde{\tau})\right].
\end{aligned}$$

Since for any pair of admissible trajectories $(\tau,\widetilde{\tau})$ that share the initial state $s_1$, we have $f_{\pi_\beta^\star}(\tau,\widetilde{\tau}) = r(\tau) - r(\widetilde{\tau})$, then $\delta^{(t)} = f_{\pi^{(t)}} - f_{\pi_\beta^\star}$ as in Equation (6).

Define $\mu^{(t)} = \frac{1}{t-1}\sum_{i<t}\pi^{(i)}\otimes\pi_{\mathsf{ref}}$, same as Xie et al. (2024), we consider a fixed step $t \geq 2$, define:

$$\mathcal{I}^{(t)} := \frac{\left(\mathbb{E}_{s_1\sim\rho,\tau\sim\pi^{(t)}|s_1,\widetilde{\tau}\sim\pi_{\mathsf{ref}}|s_1}\left[\delta^{(t)}(\tau,\widetilde{\tau})\right]\right)^2}{V_{\max}^2 \vee (t-1)\cdot\mathbb{E}_{s_1\sim\rho,(\tau,\widetilde{\tau})\sim\mu^{(t)}|s_1}\left[\left(\delta^{(t)}(\tau,\widetilde{\tau})\right)^2\right]}.$$

Using the AM-GM inequality, for any $\eta > 0$,

$$\mathbb{E}_{s_1\sim\rho,\tau\sim\pi^{(t)}|s_1,\widetilde{\tau}\sim\pi_{\mathsf{ref}}|s_1}\left[\delta^{(t)}(\tau,\widetilde{\tau})\right] \leq \frac{\mathcal{I}^{(t)}}{2\eta} + \frac{\eta}{2}\left(V_{\max}^2 + (t-1)\mathbb{E}_{s_1\sim\rho,(\tau,\widetilde{\tau})\sim\mu^{(t)}|s_1}\left[\left(\delta^{(t)}(\tau,\widetilde{\tau})\right)^2\right]\right).$$

Following Proposition 13 in Xie et al. (2022) and the definition of $\mathcal{I}^{(t)}$, we have that $\sum_{t=1}^{T} \mathcal{I}^{(t)} \leq C_{\text{cov}}(\Pi) \cdot O(\log T)$, then

$$\frac{1}{T} \sum_{t=2}^{T} \mathbb{E}_{s_1 \sim \rho, \tau \sim \pi^{(t)}|s_1, \widetilde{\tau} \sim \pi_{\text{ref}}|s_1} \left[ \delta^{(t)}(\tau, \widetilde{\tau}) \right] \leq \frac{C_{\text{cov}}(\Pi) \log T}{2\eta T} + \frac{\eta V_{\max}^2}{2}$$
$$+ \frac{\eta}{2T} \sum_{t=2}^{T} (t-1) \cdot \mathbb{E}_{s_1 \sim \rho, (\tau, \widetilde{\tau}) \sim \mu^{(t)}|s_1} \left[ \left( \delta^{(t)}(\tau, \widetilde{\tau}) \right)^2 \right].$$

Then we conclude that

$$\frac{1}{T} \sum_{t=1}^{T} J_\beta(\pi_\beta^\star) - J_\beta(\pi^{(t)}) \leq \frac{6 V_{\max}}{T} + \frac{C_{\text{cov}}(\Pi) \log T}{2\eta T} + \frac{\eta V_{\max}^2}{2}$$
$$+ \frac{\eta}{2T} \sum_{t=2}^{T} (t-1) \cdot \mathbb{E}_{s_1 \sim \rho, (\tau, \widetilde{\tau}) \sim \mu^{(t)}|s_1} \left[ \left( \delta^{(t)}(\tau, \widetilde{\tau}) \right)^2 \right]$$
$$+ \frac{1}{T} \sum_{t=2}^{T} \mathbb{E}_{\tau \sim \pi_{\text{ref}}} \left[ \beta \log \pi^{(t)}(\tau) - \beta \log \pi_\beta^\star(\tau) \right].$$

Fix $t$, and consider the term

$$\frac{\eta(t-1)}{2} \mathbb{E}_{s_1 \sim \rho, (\tau, \widetilde{\tau}) \sim \mu^{(t)}|s_1} \left[ \left( \delta^{(t)}(\tau, \widetilde{\tau}) \right)^2 \right] + \mathbb{E}_{\tau \sim \pi_{\text{ref}}} \left[ \beta \log \pi^{(t)}(\tau) - \beta \log \pi_\beta^\star(\tau) \right].$$

Recall that we have

$$\gamma \mathbb{E}_{\pi_{\text{ref}}} \left[ \log \pi^{(t)}(\tau) - \log \pi_\beta^\star(\tau) \right] + \kappa \left[ \mathbb{E}_{s_1 \sim \rho, (\tau, \widetilde{\tau}) \sim \mu^{(t)}|s_1} (f_{\pi^{(t)}} - f_{\pi_\beta^\star})^2 \right] \lesssim \frac{\text{err}}{t-1} + \frac{\gamma}{\beta} V_{\max} \sqrt{\frac{\log(|\Pi|T\delta^{-1})}{t-1}}.$$

Combine all of the terms,

$$\frac{1}{T} \sum_{t=1}^{T} \left[ J_\beta(\pi_\beta^\star) - J_\beta(\pi^{(t)}) \right] \lesssim \frac{V_{\max}}{T} + \frac{C_{\text{cov}}(\Pi) \log T}{\eta T} + \eta V_{\max}^2 + \frac{1}{T} \sum_{t=2}^{T} \left[ \frac{\beta \text{err}}{\gamma(t-1)} + V_{\max} \sqrt{\frac{\log(|\Pi|T\delta^{-1})}{t-1}} \right].$$

$\square$

**Lemma D.6.** *Under the same conditions of Theorem D.3,* err *for* `PrivXPO` *in Algorithms 2 satisfies the following bounds:*

$$\text{err} = c(\varepsilon)^2 \log(|\Pi|T\delta^{-1})$$

*where* $c(\varepsilon) := \frac{e^\varepsilon + 1}{e^\varepsilon - 1} = \frac{1}{2\sigma(\varepsilon) - 1}$.

*Proof.* Notice that Lemma 3.1 can be directly mapped to our `PrivXPO` algorithm. In particular, the parameter space $\Theta$ in the lemma corresponds to the finite policy set $\Pi$, and each parameter $\theta \in \Theta$ represents a specific policy $\pi \in \Pi$. The inputs $x^{(t)}$ are instantiated as the trajectory pairs $(\tau^{(t)}, \widetilde{\tau}^{(t)})$ generated in each round of `PrivXPO`, while the labels $y^{(t)} \in \{-1, 1\}$ represent preferences over these pairs given by BT-model. The conditional distribution $P_\theta(y|x)$ is exactly the preference probability induced by a policy $\pi$, which in our setting takes the form $P_\pi(y = 1|x^{(t)}) = \sigma(h_{\text{XPO}}^{(t)}(\pi))$. The privatized labels $\widetilde{y}^{(t)}$ are obtained through the RR mechanism and correspond to the private feedback processed by `PrivXPO`. Finally, the privatized negative log-likelihood $\widehat{L}^{(n)}(\theta)$ defined in the lemma matches precisely the empirical private loss $\widehat{L}_{\text{XPO}}^{(t)}(\pi)$ used in our algorithm. In this way, Lemma 3.1 provides a uniform convergence guarantee that underpins the generalization analysis of `PrivXPO` in the presence of local privacy.

**Lemma D.7** (Adapted from Rosset et al. (2024)). *If* $x \in [-X, X]$ *and* $y \in [-Y, Y]$, *for* $X \geq 0$, $Y \geq 1$, *then*

$$|x - y| \leq 8(X + Y)e^{2Y}|\sigma(x) - \sigma(y)|.$$

By applying Lemma D.7, we have

$$\|\delta^{(t)}(\tau, \widetilde{\tau})\| \leq 8(V_{\max} + R_{\max})e^{2R_{\max}} \cdot \|P_{\pi^{(t)}}(y = 1|\tau, \widetilde{\tau}) - P_{\pi_\beta^\star}(y = 1|\tau, \widetilde{\tau})\|.$$

Given Bernoulli distributions, we have

$$\|P_{\pi^{(t)}} - P_{\pi_\beta^\star}\|_{\mathsf{TV}}^2 = \|P_{\pi^{(t)}}(y = 1) - P_{\pi_\beta^\star}(y = 1)\|^2.$$

By defining $\kappa := \left(8(R_{\max} + V_{\max})e^{2R_{\max}}\right)^{-2}$, we derive

$$\mathbb{E}[(\delta^{(t)})^2] \lesssim \kappa^{-1} \cdot \mathbb{E}[\|P_{\pi^{(t)}} - P_{\pi_\beta^\star}\|_{\mathsf{TV}}^2]$$

By directly applying Lemma 3.1, we know that with the probability at least $1 - \delta$, for any $t \in [T]$, all $\pi \in \Pi$:

$$\sum_{i=1}^t \mathbb{E}\left[\left\|P_{\pi^{(t)}} - P_{\pi_\beta^\star}\right\|_{\mathsf{TV}}^2 |\mathcal{F}^{(i-1)}\right] \lesssim c(\varepsilon)^2 \left(\widehat{L}_{\mathsf{XPO}}^{(t)}(\pi) - \widehat{L}_{\mathsf{XPO}}^{(t)}(\pi_\beta^\star) + \log(|\Pi|\delta^{-1})\right),$$

where $c(\varepsilon) := \frac{e^\varepsilon + 1}{e^\varepsilon - 1} = \frac{1}{2\sigma(\varepsilon) - 1}$.

Thus, we obtain the implementation of Lemma D.4 under `PrivXPO`.

$$\kappa \cdot \mathbb{E}[(\delta^{(t)})^2] \lesssim c(\varepsilon)^2 \widehat{L}_{\mathsf{XPO}}^{(t)}(\pi_\beta^\star) + c(\varepsilon)^2 \widehat{L}_{\mathsf{XPO}}^{(t)}(\pi) + c(\varepsilon)^2 \log(|\Pi|T\delta^{-1}),$$

which completes the proof.

$\square$

*Proof of Theorem 4.9.* Apply the settings of Lemma D.6 to Equation (6), we have

$$\gamma\mathbb{E}_{\pi_{\mathsf{ref}}}[\log \pi^{(t)}(\tau) - \log \pi_\beta^\star(\tau)] + \kappa \cdot \mathbb{E}_{\mu^{(t)}}[(\delta^{(t)})^2] \lesssim \frac{c(\varepsilon)^2 \log(|\Pi|T\delta^{-1})}{t - 1} + \frac{\gamma}{\beta}V_{\max}\sqrt{\frac{\log(|\Pi|T\delta^{-1})}{t - 1}}.$$

Set $\eta = \frac{\beta\kappa}{\gamma T}$, and $\gamma = c(\varepsilon)\sqrt{\frac{\beta\kappa \cdot \beta \log(|\Pi|T\delta^{-1})}{T \cdot C_{\mathsf{cov}}(\Pi)}}$, we derive the final result:

$$J_\beta(\pi_\beta^\star) - J_\beta(\widehat{\pi}) \lesssim c(\varepsilon) \cdot (V_{\max} + R_{\max})e^{2R_{\max}}\sqrt{\frac{C_{\mathsf{cov}}(\Pi)\log(|\Pi|T\delta^{-1})\log^2 T}{T}},$$

where $c(\varepsilon) = \frac{e^\varepsilon + 1}{e^\varepsilon - 1}$, which completes the proof. $\square$

# E. Missing Proofs For Section 5

## E.1. Detailed Proof of Theorem 5.2

We first adapt the meta theorem of Zhou et al. (2025b, Theorem C.1) to our notation for `SquareχPO`.

**Theorem E.1.** *Suppose Assumptions 4.1 and 4.2 hold. Define $\widehat{r}(\tau) := \beta\phi\left(\frac{\widehat{\pi}(\tau)}{\pi_{\mathsf{ref}}(\tau)}\right)$ for the output $\widehat{\pi}$ of Algorithm 3. Then, we have*

$$J(\pi^\star) - J(\widehat{\pi}) \leq \frac{2V_{\max}}{R_{\max}}\sqrt{C^{\pi^\star} \cdot \mathsf{err}_{\mathsf{stat}}^2} + \beta \cdot C^{\pi^\star} + 2\beta^{-1} \cdot \frac{V_{\max}^2 \mathsf{err}_{\mathsf{stat}}^2}{R_{\max}^2},$$

*where*

$$\text{err}_{\text{stat}}^2 = \mathbb{E}_{\pi_{\text{ref}},\pi_{\text{ref}}}\left[\left(\text{clip}_{2R_{\max}}[\widehat{\Delta}(\tau,\tau')] - \text{clip}_{2R_{\max}}[\Delta^\star(\tau,\tau')]\right)^2\right],$$

*with* $\widehat{\Delta}(\tau,\tau') := \widehat{r}(\tau) - \widehat{r}(\tau')$ *and* $\Delta^\star(\tau,\tau') := r^\star(\tau) - r^\star(\tau')$. *Furthermore, by taking* $\beta = \sqrt{\frac{2}{C^{\pi^\star}}} \cdot \frac{V_{\max}\text{err}_{\text{stat}}}{R_{\max}}$, *we obtain*

$$J(\pi^\star) - J(\widehat{\pi}) \lesssim \frac{V_{\max}}{R_{\max}}\sqrt{C^{\pi^\star} \cdot \text{err}_{\text{stat}}^2}.$$

By this theorem, we only need to focus on the term of $\text{err}_{\text{stat}}^2$. In contrast to Zhou et al. (2025b), we will leverage our tighter uniform convergence lemma (i.e., Lemma 3.3) to achieve a better bound.

*Proof of Theorem 5.2.* To bound the statistical error, we essentially follow the same argument as in Zhou et al. (2025b), with the only difference in the final step by applying our new lemma, i.e., Lemma 3.3.

In particular, under Lemma 3.3, for a greedy solution $\widehat{h}$ that minimizes the empirical loss, we have

$$\mathcal{E}_{\text{CTL}}^{(n)}(\widehat{h}) \lesssim c(\varepsilon)^2 \log(|\mathcal{H}|\delta^{-1}) + n\alpha^2,$$
$$\mathcal{E}_{\text{LTC}}^{(n)}(\widehat{h}) \lesssim c(\varepsilon)^2 \log(|\mathcal{H}|\delta^{-1}) + nc(\varepsilon)^2\alpha^2.$$

Using the same mapping as in the proof of Lemma C.2 in Zhou et al. (2025b), we have that for CTL,

$$\mathbb{E}_{\pi_{\text{ref}},\pi_{\text{ref}}}\left[\left(\sigma(\text{clip}_{2R_{\max}}[\widehat{\Delta}]) - \sigma(\text{clip}_{2R_{\max}}[\Delta^\star])\right)^2\right] \lesssim c(\varepsilon)^2 \cdot \frac{\log(|\Pi|/\delta)}{n} + \alpha^2,$$

which gives a bound on $\text{err}_{\text{stat}}^2$ directly by the mean-value theorem that incurs an additional multiplicative factor of $e^{4R_{\max}}$ (cf. Lemma D.2). The same holds for LTC with the difference of $c(\varepsilon)^2\alpha^2$. This completes the proof via the meta-theorem above. $\square$

### E.2. Detailed Proof of Theorem 5.4

**Lemma E.2.** *Under the same conditions of Theorem D.3,* err *for* `SquareXPO` *in Algorithms 4 satisfies the following bound:*

$$\text{err}_{\text{CTL}} = c(\varepsilon)^2 \log(|\Pi|T\delta^{-1}) + t\alpha^2,$$
$$\text{err}_{\text{LTC}} = c(\varepsilon)^2 \log(|\Pi|T\delta^{-1}) + tc(\varepsilon)^2\alpha^2,$$

*where* $c(\varepsilon) := \frac{e^\varepsilon + 1}{e^\varepsilon - 1}$.

*Proof.* Lemma 3.3 instantiates cleanly for `SquareXPO`. In our setting, each input $x^{(t)}$ is the trajectory pair $(\tau^{(t)}, \tilde{\tau}^{(t)})$ generated at round $t$, and the clean label $y^{(t)} \in \{-1, 1\}$ is the true Bradley–Terry preference on this pair. We take the function class $\mathcal{H}$ to be the policy-indexed family $h_\pi(\tau, \tilde{\tau}) := 2\sigma(\widehat{h}_{\text{XPO}}(\pi)) - 1$, where $\widehat{h}_{\text{XPO}}(\pi) = \beta \log\left(\frac{\pi(\tau)}{\pi_{\text{ref}}(\tau)}\right) - \beta \log\left(\frac{\pi(\tilde{\tau})}{\pi_{\text{ref}}(\tilde{\tau})}\right)$ and $\sigma(\cdot)$ is the logistic function. Under the Bradley–Terry model, the regression target equals the conditional preference mean, so the Bayes-optimal element of $\mathcal{H}$ is $h^\star = h_{\pi_\beta^\star}$ with $h^\star(\tau^{(t)}, \tilde{\tau}^{(t)}) = \mathbb{E}[y^{(t)} \mid \tau^{(t)}, \tilde{\tau}^{(t)}] = 2\sigma\left(\beta \log \frac{\pi_\beta^\star(\tau^{(t)})}{\pi_{\text{ref}}(\tau^{(t)})} - \beta \log \frac{\pi_\beta^\star(\tilde{\tau}^{(t)})}{\pi_{\text{ref}}(\tilde{\tau}^{(t)})}\right) - 1$. When CTL/LTC are applied, the learner observes $z^{(t)}$ instead of $y^{(t)}$ and Lemma 3.3 prescribes the privatized square-loss $\widehat{L}_{\text{sq}}^{(n)}(h) = \sum_{t=1}^n (h(x^{(t)}) - c(\varepsilon)z^{(t)})^2$. This is exactly the empirical objective used by `SquareXPO` after substituting $h = h_\pi$, yielding $\widehat{L}_{\text{sqXPO}}^{(t)}(\pi)$.

By the previous definition of $\kappa$:

$$\begin{aligned}
\left\|\delta^{(t)}(\tau, \tilde{\tau})\right\|^2 &= \left\|\widehat{h}_{\text{xpo}}(\pi^{(t)}) - \widehat{h}_{\text{xpo}}(\pi_\beta^\star)\right\|^2 \\
&\lesssim \kappa^{-1} \cdot \left\|\sigma(\widehat{h}_{\text{xpo}}(\pi^{(t)})) - \sigma(\widehat{h}_{\text{xpo}}(\pi_\beta^\star))\right\|^2 \\
&\lesssim \frac{1}{4}\kappa^{-1}\left\|h_{\pi^{(t)}} - h^\star\right\|^2.
\end{aligned}$$

Therefore,

$$\mathbb{E}[(\delta^{(t)})^2] \lesssim \kappa^{-1}\mathbb{E}[(h_{\pi^{(t)}} - h^\star)^2].$$

Applying Lemma 3.3,

$$\sum_{i=1}^{t} \mathbb{E}\left[\left(h_{\pi^{(i)}}(x^{(i)}) - h^\star(x^{(i)})\right)^2 | \mathcal{F}^{(i-1)}\right] \lesssim \widehat{L}_{\mathsf{sq}}^{(t)}(h_\pi) - \widehat{L}_{\mathsf{sq}}^{(t)}(h^\star) + c(\varepsilon)^2 \log(|\Pi|\delta^{-1}) + t\alpha^2 \qquad \text{(CTL)}$$

$$\sum_{i=1}^{t} \mathbb{E}\left[\left(h_{\pi^{(i)}}(x^{(i)}) - h^\star(x^{(i)})\right)^2 | \mathcal{F}^{(i-1)}\right] \lesssim \widehat{L}_{\mathsf{sq}}^{(t)}(h_\pi) - \widehat{L}_{\mathsf{sq}}^{(t)}(h^\star) + c(\varepsilon)^2 \log(|\Pi|\delta^{-1}) + tc(\varepsilon)^2\alpha^2 \qquad \text{(LTC)}.$$

Thus, like before, we obtain the implementation of Lemma D.4 under `SquareXPO`.

$$\kappa \cdot \mathbb{E}[(\delta^{(t)})^2] \lesssim \widehat{L}_{\mathsf{sq}}^{(t)}(h_\pi) - \widehat{L}_{\mathsf{sq}}^{(t)}(h^\star) + c(\varepsilon)^2 \log(|\Pi|\delta^{-1}) + t\alpha^2 \qquad \text{(CTL)}$$

$$\kappa \cdot \mathbb{E}[(\delta^{(t)})^2] \lesssim \widehat{L}_{\mathsf{sq}}^{(t)}(h_\pi) - \widehat{L}_{\mathsf{sq}}^{(t)}(h^\star) + c(\varepsilon)^2 \log(|\Pi|\delta^{-1}) + tc(\varepsilon)^2\alpha^2 \qquad \text{(LTC)},$$

which completes the proof. □

*Proof of Theorem 5.4.* Apply the settings of Lemma E.2 to Equation (6), for CTL we derive:

$$\gamma\mathbb{E}_{\pi_{\mathrm{ref}}}\left[\log \pi^{(t)}(\tau) - \log \pi_\beta^\star(\tau)\right] + \kappa \cdot \mathbb{E}_{\mu^{(t)}}\left[(\delta^{(t)})^2\right] \lesssim \frac{c(\varepsilon)^2 \log(|\Pi|T\delta^{-1}) + t\alpha^2}{t-1} + \frac{\gamma}{\beta}V_{\max}\sqrt{\frac{\log(|\Pi|T\delta^{-1})}{t-1}}.$$

Set $\eta = \frac{\beta\kappa}{\alpha T}$, and $\gamma = \sqrt{\frac{\beta\kappa \cdot \beta(c(\varepsilon)^2 \log(|\Pi|T\delta^{-1}) + t\alpha^2)}{T \cdot C_{\mathsf{cov}}(\Pi)}}$, we derive the final result:

$$J_\beta(\pi_\beta^\star) - J_\beta(\widehat{\pi}_{\mathsf{CTL}}) \lesssim (V_{\max} + R_{\max})e^{2R_{\max}} \log T \sqrt{C_{\mathsf{cov}}(\Pi)}\left(c(\varepsilon) \cdot \sqrt{\frac{\log(|\Pi|T\delta^{-1})}{T}} + \alpha\right),$$

where $c(\varepsilon) = \frac{e^\varepsilon + 1}{e^\varepsilon - 1}$.

Similarly, for LTC, we have

$$J_\beta(\pi_\beta^\star) - J_\beta(\widehat{\pi}_{\mathsf{LTC}}) \lesssim (V_{\max} + R_{\max})e^{2R_{\max}} \log T \sqrt{C_{\mathsf{cov}}(\Pi)}\left(c(\varepsilon) \cdot \sqrt{\frac{\log(|\Pi|T\delta^{-1})}{T}} + c(\varepsilon)\alpha\right),$$

where $c(\varepsilon) = \frac{e^\varepsilon + 1}{e^\varepsilon - 1}$, $\eta = \frac{\beta\kappa}{\alpha T}$, and $\gamma = \sqrt{\frac{\beta\kappa \cdot \beta(c(\varepsilon)^2 \log(|\Pi|T\delta^{-1}) + tc(\varepsilon)^2\alpha^2)}{T \cdot C_{\mathsf{cov}}(\Pi)}}$.

Then we complete the proof. □

## F. Further Discussions

In this section, we discuss further applications of our two uniform convergence results in the context of alignment (e.g., reinforcement learning from human feedback (RLHF), reward model learning).

**Novelty and Conceptual Insights.** We provide additional discussion to clarify the conceptual novelty and technical insights of our results, complementing the main theoretical developments presented in the paper. A prevailing belief in the alignment literature is that MLE-type log loss cannot achieve optimal statistical rates under local privacy constraints, which has motivated the adoption of alternative objectives such as square loss. In contrast, our results demonstrate that MLE-type log loss can still attain near-optimal rates in private alignment. This is enabled by a new uniform convergence bound under log loss (Lemma 3.1), which appears to be previously unavailable in the presence of local privacy and may be of independent theoretical interest.

Our analysis also resolves an open question in prior work regarding the optimal dependence on the corruption parameter and its interplay with local privacy. By establishing a new uniform convergence bound under square loss without introducing stronger assumptions (Lemma 3.3), we obtain guarantees that simultaneously achieve optimal dependence on both privacy and corruption parameters. The key technical ingredient is a new analytical approach that fully exploits the probabilistic structure of the Huber corruption model.

**Reward model learning.** Our two uniform convergence lemmas directly give a bound for the estimation error for the greedy estimator that minimizes the empirical loss, in the context of privacy and corruption. Note that this estimation error is in terms of probability difference. To convert it to a reward difference in expectation, one can simply use the mean-value theorem and pay the inversion cost of $e^{R_{\max}}$ (e.g., Zhan et al. (2023a); Zhou et al. (2025b)). If the interest is the *empirical* reward difference (as in Zhu et al. (2023)), one can further convert from the population one to the empirical one, say using Lemma C.1 in Zhao et al. (2024) or Lemma 39 in Zanette et al. (2021) for the special case of a linear model. With such a reward model learning result, one can then leverage the reduction framework in Zhou et al. (2025a) to derive results for both RLHF and DPO.

**Improved bounds for regularized objective in (2) for large $\beta$.** For the regularized objective with a large $\beta > 0$, one can leverage the local strong convexity of the KL-divergence to achieve a faster rate of $1/(\beta T)$ rather than $1/\sqrt{T}$ (but is independent of $\beta$, hence true for any $\beta$). This has been hinted for XPO in Xie et al. (2024) and recently formally proved (although for different algorithms) in Zhao et al. (2024; 2025b;a) without privacy and corruption. In particular, Zhao et al. (2025b) gives the first faster rate in the offline alignment with single-policy concentrability. We can easily modify it to give a private and/or robust version of it. For the privacy-only case, we can simply modify their non-private MLE estimation (i.e., Algorithm 4 in Zhao et al. (2025b)) with our private one in Lemma 3.1. For the privacy-corruption cases (CTL and LTC), we can replace the objective by our square loss in Lemma 3.3. To derive the guarantees of these two new algorithms, an astute reader may already realize the same trick used in our main paper—replacing the statistical/estimation error term with our new bounds. More specifically, we only need to replace the bound in Lemma F.1 of Zhao et al. (2025b) by our new results in Lemmas 3.1 and 3.3, respectively (with mean-value theorem conversions). Then, we can have the following results:

**Proposition F.1.** *There exist offline alignment algorithms such that with probability $1 - \delta$, it holds that*

$$J_\beta(\pi_\beta^\star) - J_\beta(\widehat{\pi}_{\mathsf{Priv}}) \leq \widetilde{O}\left(\frac{1}{\beta T} D_{\pi_\beta^*}^2 \cdot c(\varepsilon)^2 \log(|\mathcal{G}|/\delta)\right)$$

$$J_\beta(\pi_\beta^\star) - J_\beta(\widehat{\pi}_{\mathsf{CTL}}) \leq \widetilde{O}\left(\frac{1}{\beta T} D_{\pi_\beta^*}^2 \cdot \left(c(\varepsilon)^2 \log(|\mathcal{G}|/\delta) + \alpha^2\right)\right)$$

$$J_\beta(\pi_\beta^\star) - J_\beta(\widehat{\pi}_{\mathsf{LTC}}) \leq \widetilde{O}\left(\frac{1}{\beta T} D_{\pi_\beta^*}^2 \cdot \left(c(\varepsilon)^2 \log(|\mathcal{G}|/\delta) + c(\varepsilon)^2 \alpha^2\right)\right),$$

*where $D_{\pi_\beta^*}^2$ is the same single-policy concentrability in Zhao et al. (2025b) and let $\mathcal{G}$ denote the policy class $\Pi$. These results reduce to the one inin Theorem D.1 of Zhao et al. (2025b) when $\varepsilon = \infty$ and $\alpha = 0$.*

Moving to the online setting, Zhao et al. (2025a) gives the first fast rate for objective (2), in the reward-based case rather than the preference-based case. However, as already mentioned by the authors, it is quite straightforward for the extension. In particular, one only needs to define a preference-based eluder dimension by introducing a baseline in their Definition 3.3 to account for the fact that in the preference-based setting, one can only learn the reward difference. A similar extension is already used in Zhao et al. (2025b) for the offline setting. Then, as in the offline case, the final guarantee scales linearly with the estimation error (in square). Hence, using our two uniform convergence results, we have the following guarantees.

**Proposition F.2.** *There exist online alignment algorithms such that with probability $1 - \delta$, it holds that*

$$J_\beta(\pi_\beta^\star) - J_\beta(\widehat{\pi}_{\mathsf{Priv}}) \leq \widetilde{O}\left(\frac{1}{\beta T} d_{\mathsf{pref,edim}}(G, T) \cdot c(\varepsilon)^2 \log(|\mathcal{G}|T/\delta)\right)$$

$$J_\beta(\pi_\beta^\star) - J_\beta(\widehat{\pi}_{\mathsf{CTL}}) \leq \widetilde{O}\left(\frac{1}{\beta T} d_{\mathsf{pref,edim}}(G, T) \cdot \left(c(\varepsilon)^2 \log(|\mathcal{G}|T/\delta) + \alpha^2\right)\right)$$

$$J_\beta(\pi_\beta^\star) - J_\beta(\widehat{\pi}_{\mathsf{LTC}}) \leq \widetilde{O}\left(\frac{1}{\beta T} d_{\mathsf{pref,edim}}(G, T) \cdot \left(c(\varepsilon)^2 \log(|\mathcal{G}|T/\delta) + c(\varepsilon)^2 \alpha^2\right)\right),$$

*where $d_{\mathsf{pref,edim}}(G, T)$ denotes the preference-based variant of the eluder dimension introduced in Definition 3.3 of Zhao et al. (2025a). and let $\mathcal{G}$ denote the policy class $\Pi$. These results reduce to those in Theorem 4.1 of Zhao et al. (2025a) as a special case when $\varepsilon = \infty$ and $\alpha = 0$.*

