# OpenReview forum: "Improved Bounds for Private and Robust Alignment"
_ICML.cc/2026/Conference — ICML 2026 regular_

### Official Review · Reviewer_Niyk · 2026-02-24

**Soundness:** 3
**Presentation:** 3
**Significance:** 3
**Originality:** 3
**Overall Recommendation:** 4
**Confidence:** 1

**Summary:**

The paper studies the sample complexity of private and robust alignment of language models. The alignment problem here is the standard DPO/RLHF setup: given pairwise preference data, find a policy that maximizes the regularized reward. This paper studied the noisy preference label setting — either from local differential privacy (randomized response) or adversarial corruption (Huber model), or both.

The main technical contribution is two uniform convergence lemmas. Lemma 3.1 shows that the standard MLE log loss under LDP already achieves near-optimal rates, overturning the prior work (Chowdhury et al., 2024b) that a custom de-biased loss is needed. Lemma 3.3 gives a tighter uniform convergence bound for square loss under both LDP and Huber corruption. These are plugged into existing alignment frameworks to obtain improved guarantees in both offline and online settings.

**Compliance With Llm Reviewing Policy:**

Affirmed.

**Final Justification:**

Overall positive results.

**Key Questions For Authors:**

See the mentioned weakness above.

**Limitations:**

yes

**Strengths And Weaknesses:**

Good:

1. The "MLE log loss is not broken under privacy" result is a clean resolution of an open question from prior work. The previous belief that MLE is suboptimal had already influenced algorithm design (de-biased losses in Zhou et al., 2025a; Chowdhury et al., 2024b), so clarifying that this was an artifact of the analysis rather than a fundamental limitation is a useful contribution to the theory community.
2. The improved corruption dependence $\sqrt{\alpha}$ to $\alpha$ is shown to be tight via MAB lower bounds, which is good.
3. The paper is relatively well-organized. The comparison tables in Appendix C make the improvement over prior work concrete.

Weakness:

I am not an expert in learning theory, so I cannot comment deeply on the technical novelty. My concerns are more on the presentation and framing side.

1. The proof intuition for the two key lemmas is entirely in the appendix. Since the main contribution is about sharper analysis, I wish the main body had a paragraph explaining where the old analysis was loose and what the new insight is.
2. The experiments are minimal — one synthetic dataset, one model, one $\epsilon$ value, no corruption. I think varying the parameters would make this section more useful.
3. The paper covers a lot of settings (two losses × two noise models × offline/online) and I found it hard to walk away with a clear takeaway. When should a practitioner use log loss vs. square loss?
4. The practical relevance is not fully clear to me. Is private RLHF (local DP on preference labels) actually deployed or being considered in practice? A sentence or two on motivation beyond the theoretical question would help.

---

> ### Author Rebuttal · Authors · 2026-03-30
>
> Thank you for your positive evaluation and your appreciation of our improvement over prior work. We will provide our response to your questions below.
>
>
> **W1. Proof intuition**
>
> We agree that the main body should better explain where the improvement comes from. The improvement in the corruption dependence comes from a new analytical perspective. Rather than applying a worst-case treatment to corrupted labels as in prior work, we explicitly exploit the probability-mixture structure of the Huber model.
>
>
>
> **W2. More experiments**
>
> We are very willing to include additional comparative experiments in the next revision to further enrich our empirical section.
> We also point out that for Square$\chi$PO, prior work [R1] has done extensive experimental studies. Our main contribution (at least for this particular algorithm) is to establish its tighter theoretical upper bounds.
>
> **W3. Square loss vs. log loss**
>
> We tend to suggest the following simple takeaway: if one needs to handle possible corruption, square loss is the safer choice; if the setting is privacy-only, then we would recommend our MLE-based log loss, since it could lead to a simple optimization. For example, in the linear model, standard MLE-log loss gives a convex loss, while square loss is non-convex.
>
>
> **W4. Practical relevance**
>
> Label privacy is relevant when preference feedback is sensitive. Even when prompts are public, the preference labels themselves can reveal user intent, values, safety preferences, or domain-specific confidential information. In particular, preference feedback may expose diagnostic reasoning, legal deliberation, or interpretive biases [R2]. We will add more motivation in our next version.
>
>
> [R1] Zhou, X., Wu, Y., Weng, W., and Orabona, F. Square$\chi$PO: Differentially Private and Robust $\chi$2-Preference Optimization in Offline Direct Alignment. International Conference on Machine Learning (ICML), 2025b
>
> [R2] Teku, N., Tian, F., Bhattacharjee, P., Chakraborty, S., Bedi, A. S., \& Tandon, R. (2025). PROPS: Progressively Private Self-alignment of Large Language Models. Transactions on Machine Learning Research. arXiv:2508.06783.

---

> > ### Author Rebuttal · Reviewer_Niyk · 2026-04-02
> >
> > My key questions are more or less addressed with the authors' promise of editing in the next version. I'd maintain my score for now.

---

> > > ### Author Response · Authors · 2026-04-05
> > >
> > > Dear Reviewer Niyk,
> > >
> > > Thanks for the acknowledgement. Following your great suggestions, we will definitely add more experiments in the next version. We appreciate your current Weak Accept support. If, during final calibration, you feel the planned edits are sufficient, we would be grateful if you could consider a modest upward adjustment; we fully respect your judgment either way.
> > >
> > > Best,
> > >
> > > Your Authors

---

### Official Review · Reviewer_Lumb · 2026-03-13

**Soundness:** 3
**Presentation:** 3
**Significance:** 3
**Originality:** 3
**Overall Recommendation:** 4
**Confidence:** 3

**Summary:**

This submission studies the theoretical sample complexity of language-model alignment from pairwise preference labels when those labels are affected by (i) label privacy and/or (ii) adversarial corruption. Concretely, the paper models preferences with the Bradley–Terry form and defines offline alignment from an i.i.d. preference dataset collected under a reference policy, versus online alignment with iterative interaction and a KL-regularized objective.

To model privacy, the paper adopts local differential privacy on the preference label—implemented via randomized response (label-flipping) with privacy parameter $\varepsilon$. To model robustness, it uses Huber contamination with corruption level $\alpha$, and distinguishes two realistic orderings of the pipeline: corruption-then-privacy (CTL) versus privacy-then-corruption (LTC), which can yield different statistical difficulty and rates.

The submission’s contributions are anchored by two new uniform convergence results under these noise models. First, it establishes a log-loss uniform convergence lemma under randomized-response LDP (Lemma 3.1), implying that a standard MLE-style log loss on privatized labels can still achieve near-optimal privacy dependence, addressing a concern in prior analyses that suggested needing specially “de-biased” losses. Second, it provides a square-loss uniform convergence lemma under CTL/LTC (Lemma 3.3) that yields improved corruption dependence (notably distinguishing $\alpha^2$-type effects and different scaling under CTL vs LTC) and sharpens earlier results.

Using these lemmas, the paper proposes (or re-analyzes) simple variants of existing alignment algorithms: an offline private algorithm (Priv$\chi$PO) and an online private algorithm (PrivXPO) based on the log-loss analysis, together with improved offline guarantees for an existing square-loss method (Square$\chi$PO) and a new online square-loss method (SquareXPO) covering joint privacy-and-corruption. The main theorems give suboptimality gap bounds where privacy typically enters via a multiplicative $c(\varepsilon)$ factor, while corruption enters additively as $\alpha$ (CTL) versus $c(\varepsilon)\alpha$ (LTC), and online bounds replace offline “concentrability” dependence with “coverability” dependence to reflect benefits of exploration.

**Compliance With Llm Reviewing Policy:**

Affirmed.

**Key Questions For Authors:**

1. Positioning vs prior online LDP-RLHF theory. How does your “online alignment with label privacy” setting and guarantee differ from prior work on online KL-regularized RLHF under $\varepsilon$-LDP (e.g., [1] that derives online regret bounds)? Please clarify differences in (a) objective (suboptimality gap vs regret), (b) interaction model / exploration assumptions, and (c) function-class complexity measures.
Why this matters: If the settings are essentially comparable, the novelty score should be reduced; if they are meaningfully different, a clearer statement would strengthen originality.

[1] Wu Y, Thareja R, Vepakomma P, et al. Offline and Online KL-Regularized RLHF under Differential Privacy[J]. arXiv preprint arXiv:2510.13512, 2025.

2. From finite $\Pi/\Theta$ to practical policy classes. Several core statements are presented for finite classes with remarks about covering-number extensions. Can you provide one explicit corollary in the main paper for a commonly used parametric class (e.g., log-linear or linear features), with the bound stated in terms of dimension/complexity? This directly affects how convincingly the theory relates to realistic approximation settings and could raise both significance and presentation scores.

3. Robustness model in the online setting. You assume an oblivious adversary for online corruption. Can you discuss whether any part of the analysis extends to adaptive corruption (even in restricted forms), or provide intuition/counterexamples for why adaptivity breaks the proof technique? If robustness fails under mild adaptivity, the robustness contribution may be narrower than it initially appears.

4. Practical optimization and computational burden. The algorithms are described as simple modifications (log-loss/square-loss objectives plus optimism terms), but in large policy classes the optimization could be non-convex and approximate. What optimization accuracy is needed for the theoretical guarantees to remain meaningful? If the bounds are highly sensitive to optimization error, the practical relevance could be materially reduced.

5. Clarifying the “log loss vs square loss” tradeoff. The paper motivates square loss for handling corruption (boundedness) and suggests empirical comparison as future work. Can you provide more guidance (even qualitative) on when you expect square loss to be preferable to log loss under privacy-only, and what failure modes (if any) you anticipate? This could improve significance and help readers interpret the methodological implications beyond rate statements.

**Limitations:**

No. The manuscript does not provide an adequate, dedicated discussion of limitations and potential negative societal impact; it mainly provides a brief “future directions” paragraph (e.g., calling for empirical studies and noting that stronger corruption models are open problems).

Possible suggestions:
 - Add a concise limitations section that explicitly lists the major modeling assumptions (Bradley–Terry preferences; label-only $\varepsilon$-LDP via randomized response; oblivious online corruption; realizability and coverage/coverability conditions) and explains how each could fail in realistic LLM alignment.

 - Clarify that “label privacy” does not address privacy of prompts/responses or other sensitive metadata, and discuss whether and how the analysis might change under central DP, aggregation, or privacy for richer observations.

**Strengths And Weaknesses:**

# Soundness (strengths)
The technical narrative is cohesive: two uniform convergence lemmas are stated explicitly with their dependence on $\varepsilon$ and $\alpha$, then instantiated into offline/online alignment via algorithmic loss choices that mirror the lemmas’ empirical objectives. The appendices include detailed proof material for the key lemmas and the main theorems, and the log-loss privacy lemma is explicitly reduced to a standard non-private log-loss uniform convergence statement by mapping privatized labels to a mixture likelihood.

On robustness, the square-loss lemma cleanly separates CTL versus LTC through different effective “approximation”/bias scalings ($\alpha$ vs $c(\varepsilon)\alpha$) and yields a sharper corruption dependence than prior analysis (the paper explicitly claims improvements over prior bounds in both CTL and LTC regimes). The paper also argues near-optimality by connecting to lower bounds for simpler problems (mean estimation / bandits) already known to separate CTL and LTC.

# Soundness (weaknesses)
The core results rely on several standard-but-strong assumptions (policy realizability, bounded density ratios / bounded implicit reward differences, and coverage-style conditions such as single-policy concentrability offline and coverability online). These are typical in theoretical RLHF/DPO analyses, but they narrow the scope in which the bounds are informative for large-scale LLM fine-tuning.

The online corruption setting assumes an oblivious adversary (the corruption distribution at each round is independent of other samples), which is materially weaker than adaptive corruptions that could react to the learner’s current policy. This is not necessarily a flaw, but it should be emphasized as a modeling limitation because robustness claims can be interpreted broadly by non-experts.

# Presentation (strengths)
The paper is well structured around three motivating questions (Q1–Q3) and ties each to an identifiable technical contribution and a corresponding theorem or lemma. The methodology is presented algorithmically, with pseudocode for the offline and online private algorithms and the online square-loss algorithm. The CTL/LTC definitions are concrete and appear early, which helps interpret the later separation results.

# Presentation (weaknesses)
The main text states finite-class results and frequently notes that extensions to infinite classes follow from covering-number arguments; however, the paper would be clearer if it provided at least one explicit corollary (e.g., a parametric log-linear or linear-feature instantiation) in the main body, rather than deferring most of the “how to instantiate” guidance to remarks/appendices.

Additionally, because the contribution is largely about rates and sharp dependence on ($\varepsilon$, $\alpha$), a short “rate summary table” (offline vs online; privacy-only vs CTL vs LTC; log loss vs square loss) would substantially improve readability and reduce the chance of misinterpreting which theorem applies to which regime.

# Significance (strengths)
Preference-label privacy and label corruption/poisoning are credible practical concerns for alignment pipelines, and the paper tackles both in a single theoretical framework spanning offline and online settings. The strongest significance claim, in my view, is conceptual: demonstrating that standard MLE-style log loss can achieve near-optimal privacy dependence under randomized response in this preference-learning context simplifies the algorithmic story (potentially reducing pressure to introduce specialized losses purely for privacy reasons).

Also, providing online guarantees under privacy and under privacy+corruption is aligned with current interest in moving beyond offline coverage limitations via exploration.

# Significance (weaknesses)
There is no empirical evaluation in this manuscript (the conclusion explicitly lists empirical comparison of log loss vs square loss as future work), which makes it hard to judge how the theoretically motivated loss choices and constants translate to practical fine-tuning dynamics for large models. This is acceptable for a theory-focused contribution, but it does limit impact on practitioner adoption unless supported by follow-up experiments.

# Originality (strengths)
Relative to prior preference-optimization theory (e.g., DPO, $\chi$PO, XPO), the novelty is primarily in (i) new uniform convergence statements under label privacy and under privacy+Huber corruption, and (ii) leveraging those to obtain sharper rates and online extensions with minimal algorithmic changes. The separation between CTL and LTC, and the emphasis that order matters, is consistent with—and technically grounded in—LDP/robust bandit literature.

# Originality (weaknesses)
Some “first” claims appear potentially overstated without discussion of closely related contemporaneous work. In particular, the manuscript claims a first sample-complexity result for online alignment with label privacy (and “first” online results appear in the abstract/introduction and around Theorem 4.8). However, there is at least one earlier public preprint [1] that studies both offline and online KL-regularized RLHF under $\varepsilon$-LDP label privacy and claims to be the first theoretical investigation of the online KL-regularized RLHF-with-LDP setting (deriving regret bounds). Even if the models/objectives/metrics differ (regret vs suboptimality gap; specific algorithmic templates), the related-work section should explicitly delineate these differences to avoid confusion and to justify novelty claims precisely.

Similarly, the paper positions itself as improving guarantees for an existing offline private+robust square-loss algorithm (Square$\chi$PO), so it should be explicit about what is genuinely new (the uniform convergence lemma and improved analysis) versus what is re-analyzed.

[1] Wu Y, Thareja R, Vepakomma P, et al. Offline and Online KL-Regularized RLHF under Differential Privacy[J]. arXiv preprint arXiv:2510.13512, 2025.

---

> ### Author Rebuttal · Authors · 2026-03-30
>
> Thank you for the positive evaluation and for the detailed questions. We address them point by point below.
>
> **Q1. Comparison with prior LDP-RLHF theory.**
>
> We have indeed cited it Wu et al [R1], in Appendix A (see line 620). We now provide more comparisons to better position our current paper.
>
> - ***Objective.*** [R1] studies only the KL-regularized objective, whereas we analyze both the unregularized and KL-regularized settings. This distinction matters because the faster KL-regularized rates require $\beta$ to be sufficiently large, so covering both large and small $\beta$ regimes is important. Moreover, our results for both unregularized and regularized settings are achieved by leveraging our two new uniform convergence results, further demonstrating the power of these lemmas.
>
> - ***Corruption or not.*** [R1] focus only on privacy. Our work provides a more robust framework by additionally studying $\alpha$-Huber corruption and its interplay with $\epsilon$-LDP. We establish the first online guarantees for two distinct interaction sequences: Corruption-then-LDP and LDP-then-Corruption. In other words, our results include theirs by setting $\alpha = 0$.
>
>
>
> **Q2. From finite classes to practical policy classes.**
>
> We agree that an explicit corollary would improve the paper. For the particular case of a log-linear policy, parametrized by $ d$-dimensional weight vector with bounded norm, we can simply replace $\log (|\Pi|)$ with $\tilde{O}(d)$ in all the theorems.
>
>
>
> **Q3. Adaptive corruption in the online setting.**
>
> We assume an oblivious adversary mainly for the ease of presentation. Our result can be directly extended to an adaptive adversary which can choose the current corruption based on history, since our analysis is based on a martingale (which can handle adaptivity) rather than i.i.d (see Appendix D.)
>
>
> **Q4. Optimization accuracy and computation.**
>
> Our theory is stated for the exact empirical optimizer for clarity, but exact global optimization is not required. As in the standard analysis, as long as the optimization error is dominated by the statistical error, we are fine.
>
> **Q5. Log loss vs. square loss.**
>
> It is worth noting that in privacy-only settings, log loss is typically preferred as it yields a more computationally tractable optimization problem. For example, under a log-linear policy, optimizing the log loss is a convex problem, but optimizing the square loss leads to a non-convex objective. However, if there exists corruption, we tend to suggest using the square loss due to its boundedness.
>
> [R1] Wu, Y., Thareja, R., Vepakomma, P., and Orabona, F. *Offline and online KL-regularized RLHF under differential privacy*, 2025.

---

> > ### Author Rebuttal · Reviewer_Lumb · 2026-04-04
> >
> > Thank the author for answering my questions, and it basically solves my concerns. I will keep my positive score.

---

> > > ### Author Response · Authors · 2026-04-05
> > >
> > > Dear Reviewer Lumb,
> > >
> > > Thank you for the acknowledgement and for confirming that your concerns are fully resolved. We appreciate your Weak Accept support. If you find it appropriate during final calibration, we would be very grateful for consideration of a one-step score increase. We fully respect your judgment either way.
> > >
> > > Best,
> > >
> > > Your Authors

---

### Official Review · Reviewer_EiqF · 2026-03-17

**Soundness:** 2
**Presentation:** 3
**Significance:** 3
**Originality:** 3
**Overall Recommendation:** 4
**Confidence:** 2

**Summary:**

This paper studies private and robust alignment from preference data in both offline and online settings, under the Bradley-Terry model. The main point on the privacy-only side is that, contrary to some prior work, a standard private MLE/log-loss objective already achieves the near-optimal privacy dependence (so a specially de-biased loss is not actually necessary).  Methodologically, the authors isolate two new uniform-convergence components: a log-loss result for labels privatized by randomized response, and a square-loss result for the joint privacy-plus-Huber-corruption setting. They then instantiate the log-loss bound in two MLE-type algorithms: Priv$\chi$PO for offline alignment, which modifies $\chi$PO only through the privatized likelihood objective, and PrivXPO for online alignment, which similarly only replaces the loss in XPO.

For the joint privacy-and-corruption setting, the paper re-analyzes the existing offline Square$\chi$PO algorithm and proposes an online analogue, SquareXPO. Here the contribution is showing that prior offline guarantees were loose, and improve the corruption terms in both CTL and LTC (roughly, from square roots-type dependence to linear).

Empirically, the paper includes a small proof-of-concept experiment on a synthetic finance-preference dataset under $\varepsilon = 0.5$, comparing MLE-type log loss, de-biased log loss, and square loss. All three achieve similar win rates, which is consistent with the theory in the privacy-only regime.

**Compliance With Llm Reviewing Policy:**

Affirmed.

**Final Justification:**

As mentioned in my rebuttal reply, my questions were not major to begin with, so I will keep my original score. However, I want to remark that I am not very familiar with the area of this paper -- it's likely that my review does not adequately examine the novelty and significance of this work -- therefore, I think it makes sense to put slightly less weight on my assessment.

**Key Questions For Authors:**

See Weakness section.

**Limitations:**

Yes.

**Strengths And Weaknesses:**

**Strength**

- I think the paper studies a reasonably well-motivated problem: understanding preference-based alignment when labels are privatized and/or corrupted. The distinction between privacy-only, CTL, and LTC settings is meaningful, and it is also a plus that the paper considers both offline and online alignment rather than only the offline case.

- The main contribution is more analytical than algorithmic, but still nontrivial. In particular, the paper gives a new uniform convergence result for log loss under randomized response, and uses it to argue that an MLE-style private log-loss objective can already achieve near-optimal privacy dependence. This seems like a useful conceptual clarification relative to prior work suggesting that one needs a specially de-biased loss. Also, the overall story is clear, that it first establishes general uniform-convergence ingredients, then instantiates them in offline and online alignment algorithms.


**Weakness**

- On a quick look, Lemma 3.3 is proven by invoking Lemma D.2, which requires a uniform sup-norm approximation
$\lVert F -G \rVert_\infty \leq \alpha_{app}$. However, it seems that the CTL proof of Lemma 3.3 shows only an *one-sided* inequality of the form $F \leq h + 2\alpha$. Could you justify a bit more on why this gives a two-sided absolute-value bound / allows invoking Lemma D.2 here?

- The experiments are limited relative to the scope of the theoretical claims. In particular, it would be nice to have more evidence about the distinctions the theory emphasizes (for example CTL vs. LTC, or behavior as $\varepsilon$ and $\alpha$ vary), especially since the paper make broad claims about improved dependence on privacy/corruption parameters.

- Presentation comments:
1. Definition 2.2 proves privacy for the mechanism RR; then the paper uses more elaborate derived mechanisms/protocols, without stating the corresponding formal privacy guarantees directly for those derived mechanisms. For clearer presentation, I suggest stating the ultimate guarantees more explicitly.

2. I recommend moving Table 3 and 4 (or a succinct version of them) to the main body, since it would make the contribution clearer.

---

> ### Author Rebuttal · Authors · 2026-03-30
>
> Thank you for your positive evaluation. We are glad that you found the problem interesting and the contributions nontrivial.
>
> **W.1 On the proof of Lemma 3.3**
>
> Thanks for your careful review. This is a typo. We can add absolute operator directly here (Eqs 4 and 5) to give two-sided inequality, due to the property of Huber corruption.
>
>
>
>
> **W2. More experiments**
>
> We are very willing to include additional comparative experiments in the next revision to further enrich our empirical section.
> We also point out that for Square$\chi$PO, prior work [R1] has done extensive experimental studies (CTL/LTC). Our main contribution (at least for this particular algorithm) is to establish its tighter theoretical upper bounds.
>
>
>
> **Presentation suggestions**
> Thanks for your great suggestions. We will move a concise version of Tables 3 and 4 to the main body. We will also clarify the end-to-end privacy guarantee of the derived protocols: since RR already preserves label privacy, any subsequent procedure maintains the same privacy level by the post-processing property of differential privacy, as long as it does not access the raw data. We will clarify this in the next version.
>
> [R1] Zhou, X., Wu, Y., Weng, W., and Orabona, F. Square$\chi$PO: Differentially Private and Robust $\chi$2-Preference Optimization in Offline Direct Alignment. International Conference on Machine Learning (ICML), 2025b.

---

> > ### Author Rebuttal · Reviewer_EiqF · 2026-04-03
> >
> > Thank you for responding to my questions. My questions were not major to begin with, so I will keep my original score.

---

> > > ### Author Response · Authors · 2026-04-05
> > >
> > > Dear Reviewer EiqF,
> > >
> > > Thank you for the acknowledgement and for confirming that your concerns are fully resolved. We appreciate your Weak Accept support. If you find it appropriate during final calibration, we would be very grateful for consideration of a one-step score increase. We fully respect your judgment either way.
> > >
> > > Best,
> > >
> > > Your Authors

---

### Official Review · Reviewer_MyPh · 2026-03-24

**Soundness:** 2
**Presentation:** 3
**Significance:** 3
**Originality:** 3
**Overall Recommendation:** 3
**Confidence:** 4

**Summary:**

Overall, this manuscript focuses on a critical challenge in large language model alignment: simultaneously handling privacy protection and adversarial label corruption in both offline and online settings. The study's central area consists of theoretical analysis for private and robust alignment, aiming to derive tight sub-optimality bounds under local differential privacy (LDP) and Huber corruption.

**Compliance With Llm Reviewing Policy:**

Affirmed.

**Key Questions For Authors:**

1. The paper focuses on randomized response for LDP; could you extend the uniform convergence results to general local differential privacy mechanisms beyond randomized response?
2. You note log loss is unbounded and thus not analyzed under Huber corruption; do you have preliminary insights or conjectures on whether private log loss can be made robust to corruption?
3. The empirical evaluation uses a small synthetic dataset; would you consider adding experiments on a real preference dataset to strengthen practical validation?
4. For the online setting, you assume an oblivious adversary; how might your bounds change under an adaptive adversary?
5. You provide faster rates for large β in Proposition G.1; could you add a brief discussion of how to set β in practice for real alignment pipelines?

**Limitations:**

Yes

**Strengths And Weaknesses:**

The paper is technically rigorous and theoretically self-consistent. All core claims are supported by complete, formal proofs built on standard learning theory tools, with all underlying assumptions clearly stated upfront. The paper is well-structured, clearly written, and logically organized for a theoretical machine learning audience.
Weaknesses:
1. The paper has critical limitations in the generalizability and empirical validation of its theoretical results. Most notably, all core theoretical guarantees rely on two overly strong, unrealistic assumptions: a finite policy class and realizability, both of which fail to hold in real-world LLM alignment with over-parameterized models.
2. The paper has critical gaps in reproducibility and clarity of contribution boundaries. It does not provide sufficient training details, hyperparameter specifications.
3. The paper’s originality is concentrated almost exclusively in its theoretical analysis, with no meaningful algorithmic innovation. All proposed algorithms are minimal one-line modifications to existing alignment frameworks (χPO, XPO), with no changes to the core training pipeline or algorithmic design.

---

> ### Author Rebuttal · Authors · 2026-03-30
>
> Thanks for your time in reviewing our paper. We will recap your valuable comments and present our detailed response.
>
> **W1. About assumptions**
>
> First, we would highlight that the finite function class assumption is mainly used for presentation simplicity. As we have remarked in Remarks 3.2 and 3.4, the results can be easily generalized to the infinite case via a covering argument. Second, regarding realizability, we want to note that it is the most common assumption in theory papers on RL, and even on supervised learning. Our bounds can also be generalized to handle the case of approximate realizability by using the approximation error term in Lemma D.2. In particular, if the target policy is only approximated by the function class, the same proof structure can handle this additional approximation term that quantifies the mismatch between the true object and its best in-class approximation. We will clarify this scope more explicitly in the revision.
>
> **W2. Training details**
>
> We highlight that we detailed our experiment setup in Appendix B. We are also happy to share the source code in the next version.
>
> **W3. One-line change to existing algorithms**
>
> We argue that this one-line change property of our proposed algorithms is a merit rather than a weakness. That is, given the existing implementations or deployment of prior algorithms like XPO and $\chi$PO, our algorithms can be easily deployed to enjoy privacy and robustness guarantees, i.e., without a complex modification on the current training pipeline.
>
> **Q1. General LDP mechanism**
>
> Thanks for the interesting question. We first remark that for our current problem, RR already achieves the best possible bounds, and moreover, RR serves as the backbone for other LDP mechanisms (See Remark 2.3).  Second, when considering alternative mechanisms, one must first determine if the mechanism preserves the structural integrity of the output (e.g., discreteness or binarity) after privatization. For instance, applying a continuous Laplace noise would yield results that are no longer meaningful within the context of the current problem.
>
> **Q2. Log loss under corruption**
>
> This is an insightful question. We tend to believe that a direct MLE approach will fail due to the unboundedness in general. For the special case of a linear model (which reduces to logistic regression), there are some existing approaches to handle this. However, both deviate from the standard MLE. In particular, [R1] takes a de-biased log-loss while [R2] uses trimmed MLE with alternating optimization. Thus, it is still unclear to us how to generalize it to the case of general function approximation.
>
> **Q3. Real-world preference datasets**
>
> We agree that stronger empirical validation would help. The current synthetic experiment is intended as a controlled sanity check for the theory. In the next version, we will add experiments on a real preference dataset, such as truthy-dpo-v0.1, Anthropic HH-RLHF, and AlpacaEval. We will also broaden the parameter sweeps to show the dependence on privacy and corruption levels predicted by the theory.
>
> **Q4. Adaptive adversary in the online setting**
>
>
> We assume an oblivious adversary mainly for the ease of presentation. Our result can be directly extended to an adaptive adversary that chooses the current corruption based on history, since our analysis is based on a martingale (which can handle adaptivity) rather than i.i.d (see Appendix D.)
>
> **Q5. Regularization parameter**
>
> One practical guideline from our theory is that the faster-rate benefit appears only when $\beta$ is not too small. Appendix G shows that if $\beta$ is on the order of $1/\sqrt{T}$, the improvement largely disappears. One possible rule of thumb is that --- the choice of $\beta$ is closely related to the sample size $T$. If the sample size is small, $\beta$ cannot be set to a small value to maintain the theoretical advantage. However, setting a small $\beta$ is acceptable when the size is large.
>
> [R1] Zhou X, Wu Y, Orabona F. A unified theoretical analysis of private and robust offline alignment: from rlhf to dpo[J]. arXiv preprint arXiv:2505.15694, 2025.
>
> [R2] Awasthi P, Das A, Kong W, et al. Trimmed maximum likelihood estimation for robust generalized linear model[J]. Advances in Neural Information Processing Systems, 2022, 35: 862-873.

---

> > ### Author Rebuttal · Reviewer_MyPh · 2026-04-06
> >
> > Critical gaps can't be resolved with minor revisions

---

> > > ### Author Response · Authors · 2026-04-06
> > >
> > > Thank you.
> > >
> > > We respectfully disagree that the remaining concerns reflect “critical gaps” in the core tenets of the paper or require a significant revision of the work.
> > >
> > > 1. The central contribution of the paper is theoretical: we establish sharper guarantees for private/robust alignment under clearly stated assumptions. We do not believe the follow-up concerns call into question the correctness or significance of these main results.
> > >
> > > 2. *On the assumption:*  The finite-class and realizability assumptions are standard assumptions used to present the cleanest theorem statements. As noted in our rebuttal, the extension to richer classes follows standard covering-number arguments, and approximate realizability is already captured through the approximation term in Lemma D.2.
> > > **These are, therefore, not flaws in the analysis.**
> > >
> > > 3. *On reproducibility/training details:* **The setup is already described in Appendix B**, and can be made more explicit in revision by surfacing the dataset/model/optimizer/hyperparameter details more prominently.
> > >
> > > 4. *On the “one-line change” concern:* We do not view this as a weakness. **The fact that privacy/robustness guarantees can be obtained via minimal changes to XPO and $\chi PO$ is precisely what makes the contribution practically meaningful.**
> > >
> > > 5. *On discussion about broader LDP mechanisms, log loss under corruption, adaptive adversary and larger real-data experiments:* These are useful directions for strengthening the paper, **but they are not prerequisites for the validity of the present contribution**.
> > > The current paper already establishes its central theoretical claims in the setting it studies.
> > >
> > > In sum, we do not think it is accurate to characterize the paper as having unresolved core-tenet issues that cannot be addressed by minor revision.

---

### Official Review · Reviewer_kopk · 2026-03-25

**Soundness:** 3
**Presentation:** 4
**Significance:** 2
**Originality:** 3
**Overall Recommendation:** 4
**Confidence:** 1

**Summary:**

This paper studies the alignment problem with differential privacy, providing new bounds for MLE-type losses, and showing a simple randomized-response-style mechanism achieves optimal rates.  The paper considers both the offline and online variant of the alignment problem. The basic idea is for users to submit randomized responses of their preference data (assumed to be {-1, 1}).  When training over these randomized labels, the authors argue (somewhat surprisingly) that no debiasing of the loss is necessary (at least theoretically). The authors provide formal sub-optimality gaps for both their online and offline algorithm variants.

**Compliance With Llm Reviewing Policy:**

Affirmed.

**Key Questions For Authors:**

See above.

**Limitations:**

See weaknesses above
* No empirical results
* Weak privacy unit (label only)
* Weak mechanism (randomized response)

**Strengths And Weaknesses:**

Strengths
* The problem is well-formalized, approach is principled, and analysis is rigorous.
* The paper is well-polished.  The remarks and implications to complement the theorem statements are helpful.
* The paper studies a few different variants of the approah (online vs. offline, CTL vs. LTC, etc.)
* The optimality results are convincing.
* The paper studies a new problem for which no / limited prior work exists.  The problem itself is one that is relevant to study under privacy.

Weaknesses
* There is no empirical evidence to to supplement the theory. It's hard to imagine the proposal to use LDP and RR would work in practice without a humongous amount of data.
* The proposed approach appears to be a basic application of RR, the fact that such a simple approach already achieves optimal rates suggest the theoretical problem considered here is not that interesting.
* The paper only considers the preference data {-1, 1} to be private, not the underlying context data associated with each label. It's not clear if this is a reasonable assumption, and not adequetly discussed.

---

> ### Author Rebuttal · Authors · 2026-03-30
>
> Thank you for the positive assessment. We address your three main concerns below.
>
> **W1. No experiment**
>
> We would like to point out that we have already included experiments in the appendix (see Appendix B) to validate our theory, due to the limited space of the main paper. In the next version, we will add more experiments on real preference datasets, such as truthy-dpo-v0.1, Anthropic HH-RLHF, and AlpacaEval. We will also broaden the parameter sweeps to show the dependence on privacy and corruption levels predicted by the theory.
>
> **W2. Randomized response**
>
> We highlight that achieving optimal rates with a simple RR mechanism is a strength rather than a limitation. We demonstrate that this standard choice is actually sufficient to reach optimal rates. While prior work has also considered RR, to our knowledge, this is the first work to achieve such a result, which is made possible by the non-trivial theoretical foundations we develop—specifically, our two uniform convergence results.
>
>
>
> **W3. Label privacy**
>
> Thanks for this sharp question. We consider label (preference) privacy as the main illustrating example to showcase our improved results when compared with prior work. In fact, via a similar analysis, we can also provide privacy for both prompt and preference in the central DP model while achieving an improved rate, when compared with prior work (i.e., Zhou et al, 2025b) [R1].
>
> [R1] Zhou, X., Wu, Y., Weng, W., and Orabona, F. Square$\chi$PO: Differentially Private and Robust $\chi$2-Preference Optimization in Offline Direct Alignment. International Conference on Machine Learning (ICML), 2025b

---

> > ### Author Rebuttal · Reviewer_kopk · 2026-04-06
> >
> > Thank you for the response, your answers are more or less what I expected, maintaining my current score.

---

### Decision · Program_Chairs · 2026-04-30

**Decision:**

Accept (regular)

**Comment:**

### Metareview

This paper studies the alignment problem for large language models under local differential privacy (LDP) and adversarial (Huber) corruption. The authors establish upper bounds on the suboptimality gap for both offline and online settings. A primary contribution is demonstrating that standard MLE-style log loss achieves near-optimal rates under LDP. Furthermore, the paper provides new uniform convergence results for square loss, yielding tighter sample complexity bounds when privacy and corruption are combined (differentiating between corruption-then-privacy and privacy-then-corruption settings). Finally, the authors extend these guarantees to online alignment with active exploration.

**Strengths**

* **Elegant Algorithms (Reviewers EiqF, Lumb, Niyk):** The paper elegantly shows that standard MLE log loss still works under local privacy, removing the necessity for specially designed de-biased losses.

* **Improved Theoretical Bounds (Reviewers kopk, MyPh, Lumb):** The theoretical framework is sound, providing sharper corruption dependencies (improving from $\sqrt{\alpha}$ to $\alpha$).

* **Comprehensive Scope (Reviewers kopk, EiqF):** The paper captures both the offline and online settings, and considers both privacy-only and joint privacy-and-corruption scenarios. For the latter, both different noise model orderings (CTL vs. LTC) are considered.

**Weaknesses**

* **Algorithmic Novelty vs. Simplicity (Reviewers kopk, MyPh):** The proposed algorithms are essentially minor (one-line) modifications to existing frameworks like $\chi PO$ and XPO. While some view this as lacking algorithmic innovation, the authors argue that it is a strength.

* **Strong Theoretical Assumptions (Reviewers MyPh, Lumb):** The core guarantees rely heavily on assumptions such as finite policy classes, exact realizability, bounded density ratios, and oblivious adversaries in the online setting. For this point, the authors point out that some of the assumptions can be relaxed without too much effort, and that some of these are standard in the area.

* **Limited Empirical Validation (All reviewers):** The experimental evidence is limited, relying on a small synthetic dataset and a potentially weak benchmark, and is not even included in the main body of the paper.

** Recommendation: ** The paper studies a natural problem of alignment under DP and shows that the MLE-style algorithm for log loss already achieves near optimal rate, a very nice contribution. Even though the proofs might use standard techniques, the paper is still appropriate for a (weak) accept.